# Quantification of capillary rise dynamics in snow using neutron radiography

Michael Lombardo[1], Amelie Fees[1], Anders Kaestner[2], Alec van Herwijnen[1], Jürg Schweizer[1], and Peter Lehmann[3]

[1]WSL Institute for Snow and Avalanche Research SLF, Davos, Switzerland
[2]Laboratory for Neutron Scattering and Imaging, Paul Scherrer Institute, Villigen, Switzerland
[3]Physics of Soils and Terrestrial Ecosystems, ETH Zurich, Zurich, Switzerland

**Correspondence:** Michael Lombardo (michael.lombardo@slf.ch)

**Abstract.** Liquid water flow in snow is important for snow hydrology, remote sensing, and avalanche formation. Water flow in snow is often dominated by capillary effects, which are responsible for the formation of capillary barriers, capillary flow paths, and capillary rise. Unfortunately, there is little quantitative data on the capillary forces of snow, particularly with respect to capillary rise dynamics. Here, we present the results of 4 capillary rise experiments using neutron radiography. The experiments were performed in $13 \times 13 \times 1$ cm³ glass columns with sand-snow and sand-gravel-snow layering mimicking the capillary forces at the soil-snow interface. Images were taken at 10 s to 15 s intervals with a pixel size of 92 $\mu$m. The experiments provided quantitative results of high resolution liquid water profiles, wetting front progression, flow rates, and parameterization of snow hydraulic properties. The experiments showed that the snow properties influenced the capillary rise height while the hydraulic properties of the transitional layer below the snow influenced the flow rates. The saturated hydraulic conductivity values obtained from the experiments were below the expected values from the literature.

## 1 Introduction

The dynamics of liquid water flow in snow are important for snow hydrology (e.g., Quéno et al., 2020; Webb et al., 2020), remote sensing (e.g., Marin et al., 2020; Rösel et al., 2021), and avalanche forecasting (e.g., Wever et al., 2017; Hendrick et al., 2023). For example, snowmelt and rain-on-snow events are important for stream-flow predictions (e.g., Würzer et al., 2017; Hammond and Kampf, 2020), while the first wetting of weak layers within the snowpack can lead to an increase in avalanche activity (Wever et al., 2018). In these examples, water is generated at the snow surface through melting or rain-on-snow events and subsequently percolates towards the ground. This percolation is known to be affected by capillary forces within the snowpack, through the formation of preferential flow paths (Hirashima et al., 2019) and capillary barriers (Quéno et al., 2020). In addition to top-down percolation, capillary rise has been shown to be relevant for the wicking of brine into snow on sea ice (Mallett et al., 2024) and the formation of wet basal layers under gliding snowpacks (Mitterer and Schweizer, 2012; Lombardo et al., 2025b).

The inclusion of capillary forces in snowpack models has been shown to improve the simulation results of water flow in snow (Wever et al., 2014; D'Amboise et al., 2017). For example, 2D simulations were able to generate preferential (Leroux

et al., 2020) and lateral flow paths (Webb et al., 2018), while 1D implementations reproduced the formation of capillary barriers (Hirashima et al., 2010) and ice lenses (Quéno et al., 2020). Additionally, implementation of capillary forces has been shown to improve the estimates of snowpack runoff at sub-daily timescales (Wever et al., 2014) as well as reproduce capillary rise on sea ice (Wever et al., 2020) and at the bottom of the snowpack (Mitterer and Schweizer, 2012).

The implementation of capillary forces in snowpack models is typically done via the Richards equation (Richards, 1931; Richards and Gardner, 1936), which requires a relationship between the water pressure in the unsaturated porous medium and liquid water content, called the water retention curve, and the hydraulic conductivity function (Wever et al., 2014; D'Amboise et al., 2017; Webb et al., 2018; Leroux et al., 2020). These relationships are most commonly described for snow with the model of Mualem-van Genuchten (Mualem, 1976; van Genuchten, 1980). However, the experimental data needed for determining the parameter values of the hydraulic functions of this model are rather limited. The water retention curve of snow, for example, has only been measured directly in limited cases, typically with a vertical resolution on the order of centimeters (Colbeck, 1974, 1975; Wankiewicz, 1976, 1978; Jordan, 1983; Marsh, 1991; Coléou et al., 1999; Yamaguchi et al., 2010, 2012; Katsushima et al., 2013; Adachi et al., 2020). The most extensive study was performed by Yamaguchi et al. (2012), whose parametrization for the Mualem-van Genuchten model is the most widely used today and is based on drainage experiments of sieved snow with a vertical resolution of 2 cm. Due to the relatively large pores in snow, there are large differences between the wetting and drying branches of the water retention curves (Adachi et al., 2020). The only direct measurements of the wetting branch were by Coléou et al. (1999) and Adachi et al. (2020), with the experiments by Adachi et al. (2020) having a vertical resolution of 2 mm using magnetic resonance imaging (MRI). Other measurements of capillary rise height also exist, but are insufficient for describing the water retention curve (Coléou and Lesaffre, 1998; Marsh, 1991; Jordan, 1995). In all cases, the measurements of the water retention curves and capillary rise heights were done in hydrostatic equilibrium without capturing the flow dynamics.

While the water retention curve describes the equilibrium state, the water flow dynamics are controlled by the capillary forces and the hydraulic conductivity function. Typically, the hydraulic conductivity is measured under saturated conditions and its dependence on water content is related to the water retention curve. As with the water retention curve data, the data on the saturated hydraulic conductivity of snow are also limited. The saturated hydraulic conductivity of snow has only been directly measured in a few very limited cases (Katsushima et al., 2013; Walter et al., 2013; Clerx et al., 2022). More commonly, the saturated hydraulic conductivity is determined via the permeability, where the conductivity is the product of permeability and fluidity (a function of the liquid density and viscosity). The permeability of snow has been measured (Shimizu, 1970; Jordan et al., 1999; Albert et al., 2000) and also calculated from 3D images of snow microstructure (Courville et al., 2010; Calonne et al., 2012). The parameterization by Calonne et al. (2012) is the most commonly used method.

When considering the available data above, there are only limited measurements of capillary rise in snow, and, to the best of our knowledge, none that capture the dynamics. Here, we present the results of 4 capillary rise experiments in snow using neutron radiography. The experiments provided the 2D evolution of the capillary rise process at a spatial resolution of 92 $\mu$m and temporal resolution of 10 s to 15 s. Motivated by the role of capillary rise in the formation of glide-snow avalanches (Mitterer and Schweizer, 2012; Lombardo et al., 2025b), the experiments were performed with sand-snow and sand-gravel-

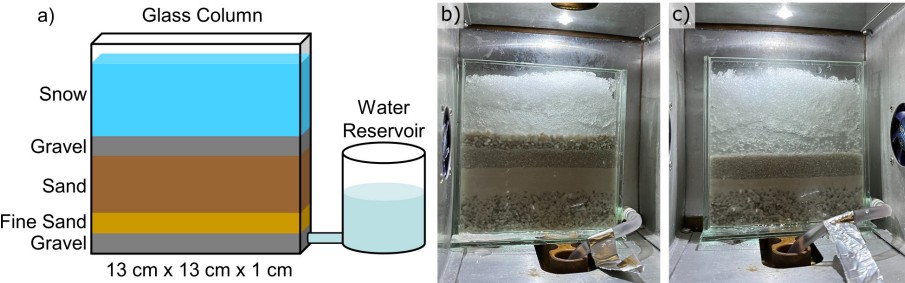

**Figure 1.** Schematic (a) and photos (b and c) of the column setup. The schematic shows a layering with the transitional gravel layer as does the photo in (b). The photo in (c) is for a layering without gravel. The gravel layer reproduces the effect of the vegetation layer found on grassy slopes. The lower boundary of the column is filled with coarse gravel to facilitate rapid distribution of water from the reservoir.

snow systems, which replicated the capillary forces at the soil-snow interface. The gravel mimicked the weak capillary forces of a vegetation layer found on grassy slopes prone to glide-snow avalanche activity (Feistl et al., 2014; Fees et al., 2025). The results show the effect of the snow and transitional layer properties on the capillary rise process.

## 2 Materials and methods

### 2.1 Neutron radiography

Neutron radiography has been shown to be one of the few measurement techniques capable of providing high-resolution, non-destructive measurements of water transport in soil-snow systems and was therefore selected for these experiments (Lombardo et al., 2025c). Neutron radiography experiments were performed on the NEUTRA beamline at the Swiss Spallation Neutron Source (SINQ) of the Paul Scherrer Institute (PSI) in Villigen, Switzerland (Lehmann et al., 2001). Glass columns filled with sand, gravel, and snow were placed inside a climatic chamber within the neutron beam (Mannes et al., 2017). An Andor iKon-L camera using a Zeiss Otus 55 mm f/1.4 lens with a 100 $\mu$m $^6$Li ZnS:Cu scintillator (RC Tritec AG, Teufen, Switzerland) was used as the detector. The resulting pixel size was 92 $\mu$m. Images were taken with 10 s or 15 s exposure depending on the experiment with an additional 3 s between exposures for data acquisition by the camera. Vertical bars of perfect neutron absorbers (black bodies made of $^{10}B_4C$) were placed in front of the climatic chamber and were used for the scattering correction described below.

### 2.2 Sample preparation

The samples placed into the beam were prepared in glass columns with face dimensions (perpendicular to the beam) of 13 cm × 13 cm with a sample thickness in the beam direction of 1 cm with 3 mm thick walls (Fig. 1a). Each column consisted of a layering of sands (Carlo Bernasconi AG (Bern, Switzerland)) and snow. Sand (0.3–0.9 mm diameter) and gravel (2.0–3.2 mm

diameter) were used as soil and vegetation analogs, respectively. These materials were used to mimic the capillary forces of the soil-vegetation-snow layering found in nature under many gliding snowpacks (Feistl et al., 2014). The gravel was used to mimic the weak capillary forces of a high porosity vegetation layer. Fine sand (0.06–0.25 mm diameter) and gravel were used at the bottom of each column to form a barrier for air infiltration from above after initial wetting, and allow for homogeneous and fast capillary rise into the soil analog layer. The sands and gravel were dry when packing the column. Each column was connected to a water reservoir filled with water at 0 °C with which the height of the water table, and thereby the pressure, was controlled.

Two snow types were used. The fine-grained (FG) snow was a natural snow sample of melt forms and small rounded grains with a grain size of 0.25–0.75 mm (Fierz et al., 2009). The coarse-grained (CG) snow was an artificial snow sample grown using the Snowmaker (Schleef et al., 2014) and stored at 0 °C with periods at +2 °C in the laboratory to generate melt forms with grain size 0.5–1.5 mm (Fierz et al., 2009). The fine-grained snow had a density of 531 kg m$^{-3}$ and coarse-grained snow had a density of 547 kg m$^{-3}$ as determined from micro-computed tomography ($\mu$CT) scans (Fig. A1). The densities were expected to vary from the laboratory values after packing into the column at PSI. Specific surface area (SSA, surface area per mass snow) was also determined from the $\mu$CT scans and was used to calculate the optically equivalent grain size needed for the parameterization described by Calonne et al. (2012). The optically equivalent grain diameter was determined using the relationship $d = 6 \times SSA^{-1} \times \rho^{-1}$ (Calonne et al., 2012). Micro-computed tomography was performed on a $\mu$CT 90 by SCANCO Medical AG (Brüttisellen, Switzerland) with a voxel size of 11 $\mu$m, and X-ray tube settings of 55 kVp and 8 W. The segmentation was performed using the method described by Hagenmuller et al. (2013).

A total of 4 samples were prepared. Two configurations (FG$_{s,g}$ and CG$_{s,g}$) had a gravel layer between the sand and the snow (Fig. 1b), while the other two (FG$_s$ and CG$_s$) had snow directly in contact with the sand (Fig. 1c). The configurations are notated with FG or CG for the snow type, with the subscript "s" denoting the sand-snow layering and "s,g" denoting the sand-gravel-snow layering.

## 2.3 Capillary rise experiment procedure

The sand and gravel layers were dry packed into the column at room temperature, after which the column was placed into an ice bath at 0 °C. During this time, the necessary neutron images (open beam and dark current, see below) without the column were acquired prior to the experiment. Once the column reached 0 °C, snow stored at -15 °C to -20 °C was transferred into the column with a spatula. The sintered snow in the storage container was broken with the spatula and poured/spooned into the container as loose grains. The column remained in the ice bath during snow packing while the ambient air temperature was room temperature (approximately 20-25 °C). The column was transferred to the climate chamber (-5 °C) in the ice bath, minimizing the time at room temperature. When the column was in the chamber, a hose was routed through the bottom of the chamber and connected to the reservoir filled with ice water. The water level was set to 1 cm below the top of the sand (soil analog) accounting for the water level loss due to the volume of water in the tubing. The hydraulic head therefore reduced slightly (approximately 0.5 cm) during the filling of the tubing and column, but is assumed to have a minimal effect on the dynamics. The chamber temperature was then set to 0 °C for the duration of the experiment. The experiment was started by

opening a valve allowing for water to flow from the reservoir into the column. Due to the safety requirements of the beamline, there was always a delay between the opening of the valve and the first image. This is why the region of interest in the first images are not completely dry and, in some experiments, the water was already within the snow by the time the first image was taken (e.g., FG$_s$). This procedural limitation has several implications on the analysis, which are discussed in more detail below.

## 2.4 Image processing

Image processing was performed using Python 3.8 with standard Python libraries. Image processing was necessary to correct the raw images for various effects such as dark-current (detector signal while the neutron beam is off), attenuation from the non-sample components such as the climate chamber, neutron scattering effects of the sample and non-sample components, and fluctuations of the neutron beam. The corrections were performed following the method described by Lombardo et al. (2025c) to obtain the unitless optical density (OD)

$$OD_n = -\ln\left( \frac{I_{n,BB} - I_{DC} - I_{n,BB}^S}{I_{OB,BB} - I_{DC} - I_{OB,BB}^S} \cdot \frac{D_{OB,BB}^p}{D_{n,BB}^p} \right) \tag{1}$$

where $OD_n$ is the optical density of the n$^{th}$ sample image, $I_{n,BB}$ is the measured intensity of the n$^{th}$ image with black bodies, $I_{DC}$ is the dark-current, $I_{OB,BB}$ is the open-beam image (empty climate chamber) with black-body bars, $I_{n,BB}^S$ is the scattering contribution of the n$^{th}$ image with black-body bars, $I_{OB,BB}^S$ is the scattering contribution of the open-beam image with black-body bars, $D_{OB,BB}^p$ is the proton dosis of the open-beam images with black-body bars, and $D_{n,BB}^p$ is the proton dosis of the n$^{th}$ image with black-body bars. The proton dosis terms correspond to the proton current at the neutron generating target and are proportional to the neutron flux.

One improvement to the method by Lombardo et al. (2025c) was the use of continuous vertical black-body bars instead of a grid of black-body cylinders to improve the vertical resolution of the scattering correction. Virtual black bodies were created along each black-body bar to replicate the cylinder grid configuration, but with a higher vertical resolution. 20 evenly-spaced virtual black bodies were placed on each bar (Fig. 2). The black-body values were calculated as the mean of an 11 pixel $\times$ 11 pixel square around each virtual black-body location. This resulted in one intensity value per virtual black body, which was subsequently used for the thin-plate spline interpolation to obtain the scattering images ($I_{n,BB}^S$ and $I_{OB,BB}^S$) (Carminati et al., 2019; Lombardo et al., 2025c). Due to significant amount of scattering from the water and ice, a beam hardening correction was applied to the scattering-corrected optical density using the polynomial expression ($y = 0.99x + 0.03x^2 + 0.0009x^3$) reported by Carminati et al. (2019) for the NEUTRA beamline, where $x$ is the calculated optical density from Eq. 1 and $y$ is the optical density after the beam-hardening correction. Finally, the optical density of the glass column walls ($0.192 \pm 0.009$) was subtracted from the beam-hardening corrected optical density. The glass value was experimentally determined using the same method described above for an empty column. The value was assumed to be the same for all columns.

The correction method was validated by measuring the linear attenuation coefficient of water and comparing it to the value of 3.6 cm$^{-1}$ reported by Carminati et al. (2019). The linear attenuation coefficient is related to the optical density through

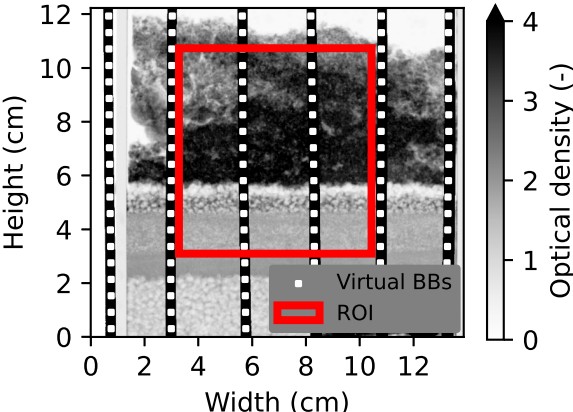

**Figure 2.** An example of an optical density image showing the region of interest (ROI) used for the calculations (red box). The black bodies are the vertical black bars and the virtual black bodies (BBs, white rectangles) were placed within the black bodies at even intervals.

$$OD = \mu z \tag{2}$$

where $OD$ is the unitless optical density, $\mu$ is the linear attenuation coefficient in cm$^{-1}$, and $z$ is the path length in cm. The linear attenuation coefficient of water was measured with an aluminum wedge with steps of different water thicknesses in the beam direction (path lengths) between 0.5 mm and 5 mm. Applied to the wedge, the correction method resulted in a linear attenuation coefficient for water of $3.60 \pm 0.02$ cm$^{-1}$ (Fig. 3), which matches the value of 3.6 cm$^{-1}$ reported by Carminati et al. (2019).

### 2.5 Liquid water content

Liquid water content, expressed as volume of water per total sample volume, of snow was derived from the optical density through the following relationships. First, the total optical density can be expressed as the sum of optical densities of the sample components (ice and water) as

$$OD_{\text{total}} = OD_{\text{ice}} + OD_{\text{water}} \tag{3}$$

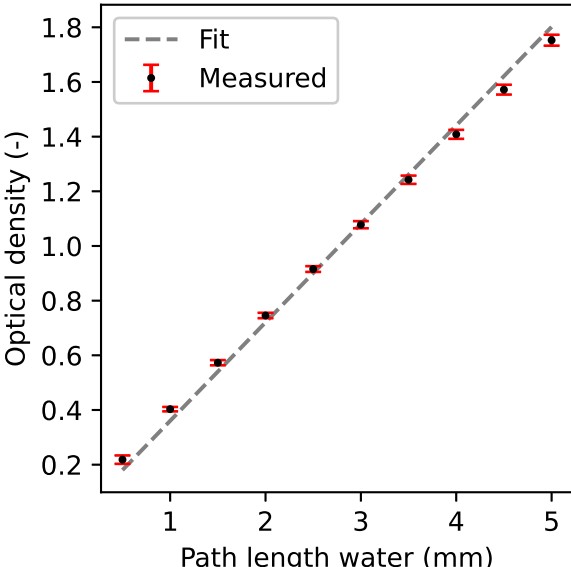

**Figure 3.** The measured optical density of the reference (aluminum wedge with varying water thicknesses) as a function of water thickness or path length (black dots) with error bars of the standard deviation (red) and linear fit (dotted gray line). Fit equation: $OD = 0.360(\pm0.002)z$, where $z$ is the path length in mm and $OD$ is the unitless optical density. The slope of 0.36 mm$^{-1}$ (or 3.6 cm$^{-1}$) is the expected attenuation coefficient of water.

where $OD_\text{total}$ is the total optical density, $OD_\text{ice}$ is the optical density of the ice, and $OD_\text{water}$ is the optical density of the
155  water. The optical density of air is negligible and the contribution of the glass was already subtracted (as explained above). Then, since optical density is the product of the linear attenuation coefficient ($\mu$) and path length ($z$), Eq. 3 can be rewritten as

$$OD_\text{total} = \mu_\text{ice}z_\text{ice} + \mu_\text{water}z_\text{water} \tag{4}$$

where the linear attenuation coefficients and path lengths are given for ice and water. Using the measured value for $\mu_\text{water}$ of $3.60 \pm 0.02$ cm$^{-1}$, the linear attenuation of ice is calculated as

160  $$\mu_\text{ice} = \mu_\text{water} \frac{\rho_\text{ice}}{\rho_\text{water}} \tag{5}$$

resulting in a value of $3.26 \pm 0.02$ cm$^{-1}$ when $\rho_\text{water}$=1000 kg m$^{-3}$ and $\rho_\text{ice}$=917 kg m$^{-3}$. This assumes that the only difference in attenuation between ice and water is the density. While the linear attenuation coefficient is also a function of neutron energy, the energy-related differences between ice and water are negligible for the thermal neutron energy spectrum (mean energy of 25 meV) of the NEUTRA beamline (Lehmann et al., 2001). Now, the path lengths of ice and water ($z_\text{ice}$ and $z_\text{water}$) can be

expressed as fractions of the total path length ($z$), which are simply the ice volume fraction and volumetric liquid water content ($\theta$) for ice and water, respectively. Thus, Eq. 4 can be rewritten as

$$OD_{\text{total}} = \mu_{\text{ice}} \frac{\rho_{\text{snow,dry}}}{\rho_{\text{ice}}} z_{\text{total}} + \mu_{\text{water}} \theta z_{\text{total}} \tag{6}$$

where $\rho_{\text{snow,dry}}$ is the dry snow density, $\theta$ is the volumetric liquid water content (given as a fraction) of the snow, and $z_{\text{total}}$ is the total path length of the column (here, 1 cm). Reorganization of Eq. 6 provides an expression for snow liquid water content

$$\theta = \frac{1}{\mu_{\text{water}}} \left( \frac{OD_{\text{total}}}{z_{\text{total}}} - \mu_{\text{ice}} \frac{\rho_{\text{snow,dry}}}{\rho_{\text{ice}}} \right) \tag{7}$$

where all quantities are known or measured except for $\rho_{\text{snow,dry}}$. Since the linear attenuation coefficients between ice and water only differ due to the difference in their densities for the NEUTRA energy spectrum, we cannot directly distinguish between ice and water when the corresponding path lengths are unknown. Thus, when calculating the liquid water content, the dry snow density must be known (described in the next section).

## 2.6   Dry snow density

The dry snow density ($\rho_{\text{snow,dry}}$) was calculated as

$$\rho_{\text{snow,dry}} = \frac{OD_{\text{dry}}}{\mu_{\text{ice}} z_{\text{total}}} \rho_{\text{ice}} \tag{8}$$

where $OD_{\text{dry}}$ is the optical density of the snow in a dry state and the other variables are defined above. This assumes that 1) the dry state was truly dry and 2) no changes to the snow occurred during the experiment (e.g., densification). The dry state was always determined from the first image. For the horizontally averaged vertical profiles, the average optical density of a manually selected vertical region of the sample (e.g., from 0.5 cm to 2 cm above the bottom of the snow layer) was taken as the dry snow density. Thus, a single dry snow density was used for the entire vertical profile. This manual selection was necessary because $CG_{s,g}$, $CG_s$, and $FG_s$ already had some water present within the snow in the first image. The dry snow density used for the vertical profiles is provided in Tab. 1. For the analysis of the 2D images, the dry state was defined on a per-pixel basis, where the dry snow density was calculated from the first image, regardless of the presence of water. The implications of these assumptions are addressed in the Discussion section.

## 2.7   Flow rates

The flow rate of water from the sand or gravel into the snow was calculated from the horizontally averaged liquid water content profiles as

$$q = \frac{\Delta \theta}{\Delta t} H \tag{9}$$

where $q$ is the flow rate in cm min$^{-1}$, $\Delta\theta$ is the change in liquid water content of the snow between two images, $\Delta t$ is the change in time (min) between the two images, and $H$ is the height (cm) of the snow within the ROI. For this calculation, the flow was assumed to be only vertical. The horizontal averaging of the liquid water content profiles was done for each pixel height. The results were smoothed with a Savitzky-Golay filter (Savitzky and Golay, 1964) with a window length of 10 and a third order polynomial to reduce the noise of the signal and make the curves more legible.

## 2.8 Water retention curve fitting

The water retention curve describes the relationship between capillary pressure and liquid water content. Here, we used the van Genuchten model (van Genuchten, 1980) given by

$$\theta\left(h\right)=\theta_{\mathrm{r}}+\frac{(\theta_{\mathrm{s}}-\theta_{\mathrm{r}})}{[1+(\alpha h)^{n}]^{m}} \tag{10}$$

where $h$ is pressure head or matric potential head (expressed here as the absolute value of the negative head or height above the water table), $\theta$ is volumetric liquid water content, $\theta_{\mathrm{r}}$ is residual liquid water content, $\theta_{\mathrm{s}}$ is saturated liquid water content, and three parameters: $\alpha$, $n$, and $m$. While $\theta_{\mathrm{r}}$ and $\theta_{\mathrm{s}}$ have direct physical meanings as the minimum and maximum amount of water, $n$ and $\alpha$ are related to the width of pore-size distribution and the inverse of the air-entry pressure, respectively (van Lier and Pinheiro, 2018).

Fitting of the water retention curves was performed on the horizontally averaged liquid water content profiles from the last image of each experiment, which assumes that the water distribution is close to hydrostatic equilibrium. An example fit of FG$_{\mathrm{s,g}}$ is provided in Fig. 4 with the remaining fits in the Appendix (Fig. A5, Fig. A6, Fig. A7). The pressures were calculated based on the height above the water level in the reservoir. The fit was performed using a nonlinear least squares approach with Eq. 10. All parameters were allowed to vary freely except $\theta_{\mathrm{r}}$, which was limited to a value close to zero (maximum 0.001). Typically, a value of around 0.02 is used for the residual water content of snow (Yamaguchi et al., 2012; Wever et al., 2014). However, since the initial wetting curve of a truly dry sample cannot start at a value greater than 0, a value of 0 was more appropriate for these experiments. The relationship $m=1-1/n$ was used (van Genuchten, 1980), where $n$ must be larger than 1.

The mean absolute error (MAE) was used to quantify the quality of the fitting as

$$MAE=\frac{1}{n_{\mathrm{m}}}\sum\left|y_{\mathrm{measured}}-y_{\mathrm{fit}}\right| \tag{11}$$

where $n_{\mathrm{m}}$ is the number of measurements, $y_{\mathrm{measured}}$ are the measured values, and $y_{\mathrm{fit}}$ are the fit values. The measured values were the horizontally averaged liquid water content values measured by neutron radiography and the fit values were from the least squares regression. The units of the MAE are liquid water content expressed as volume fraction.

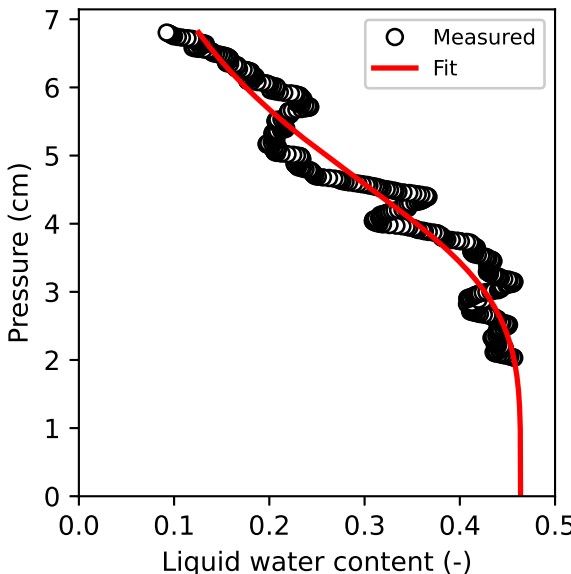

**Figure 4.** The 1D liquid water content of $FG_{s,g}$ as a function of pressure head (black circles) and the fit of these points to Eq. 10 (red line) with mean absolute error equal to 0.022 and expressed as the volume fraction of liquid water content.

## 2.9 Hydraulic conductivity fitting

Fitting of the hydraulic conductivity was performed using the inverse solution method of Hydrus-1D on the measured capillary rise data (Šimůnek et al., 2008). The fit was performed on the Mualem-van Genuchten function (Mualem, 1976; van Genuchten, 1980) which combines Eq. 10 with

$$K\left(S_{e}\right) = K_{s} S_{e}^{\frac{1}{2}} \left[1 - \left(1 - S_{e}^{\frac{1}{m}}\right)^{m}\right]^{2} \tag{12}$$

where $K_{s}$ is the hydraulic conductivity at saturation, $m$ is again taken as $1 - 1/n$, and $S_{e}$ is the effective saturation defined as

$$S_{e} = \frac{\theta\left(h\right) - \theta_{r}}{\theta_{s} - \theta_{r}} \tag{13}$$

which accounts for the pore volume occupied by the residual water. The exponent of 1/2 is the default value of the $\tau$ parameter, which describes the effects of the connectivity and tortuosity of the flow paths. This parameter was not fit in this study (see Discussion). An example fit of $FG_{s,g}$ is provided in Fig. 5 with the remaining fits in the Appendix (Fig. A8, Fig. A9, Fig. A10).

The simulations included 5 cm of snow and 0.5 cm of a transitional layer below the snow with a total of 221 numerical nodes

(spatial resolution of 0.025 cm). The transitional layer captured the upper part of the gravel layer for $FG_{s,g}$ and $CG_{s,g}$, the sand for $FG_{s}$ and $CG_{s}$, and the gravel-snow or sand-snow interface, respectively. The snow layers were initialized using the liquid

water content profile within the snow (calculated with Eq. 6 and Eq. 8). The transitional layer was initialized in a dry state (pressure $\leq$ 50 cm) with the pressure of the lowest node set to the height above the water level in the reservoir. The simulation began at t=0.

The upper boundary condition was a constant zero flux condition and the lower boundary condition was a constant pressure head based on the distance of the lower boundary to the height of the water level in the reservoir. Fitting was performed to 10 numerical nodes (called "observation nodes" in Hydrus) that act as virtual sensors to record the liquid water content. The observation nodes had a vertical spacing of 0.5 cm and fitting was performed at 11 timesteps. The parameters $\theta_r, \theta_s$, $\alpha$, $n$ of the snow were fixed to the values obtained by the water retention curve fitting and the saturated hydraulic conductivity was

allowed to vary. For $CG_{s,g}$, the static fitting resulted in a value of $n$=18.6, which made the inverse fitting difficult, as a high $n$ value leads to large changes in LWC for small changes in pressure. This prohibited the model from converging and a smaller value of 10 was needed to perform the fitting for this experiment. The transition layers in $FG_{s,g}$ and $CG_{s,g}$ were described with arbitrary values for $\theta_r$, $\theta_s$, $\alpha$, and $n$ of 0.00, 0.43, 0.5 cm$^{-1}$, and 3.0, respectively, such that the layer remained relatively dry (see Results). The $\theta_r, \theta_s$, $\alpha$, $n$ values for the transitional layers in $FG_s$ and $CG_s$ were 0.04, 0.38, 0.06 cm$^{-1}$, and 8.1, respectively,

which are the values fit to the water retention curve from drainage experiments measured in the lab for the sand (Lombardo et al., 2025c). The saturated hydraulic conductivity of the transitional layer was also fit. The MAE (Eq. 11) was calculated in the same way as for the water retention curves described above with the measured values coming from the (horizontally averaged) neutron radiography experiments and the fit values coming from the inverse fitting. The Hydrus files for each of the simulations are provided by Lombardo et al. (2025a).

**2.10   Wetting front tracking**

The wetting front was defined as the region of 50% saturation, where saturation is defined as the fraction of the pore space filled with water. The saturation ($S$) was calculated using the relationship

$$\theta = \left(1 - \frac{\rho_{\text{snow,dry}}}{\rho_{\text{ice}}}\right) S \tag{14}$$

where $1 - \frac{\rho_{\text{snow,dry}}}{\rho_{\text{ice}}}$ is the porosity, $\theta$ is the liquid water content calculated from Eq. 7, $\rho_{\text{snow,dry}}$ is the dry snow density

calculated with Eq. 8, and $\rho_{\text{ice}}$ is the density of ice (917 kg m$^{-3}$). The wetting front was tracked individually for each of the three regions separated by the black-body bars, called Left, Middle, and Right (Fig. 9). The dry density was calculated for each pixel from the first image as described above. The saturation image (for each region) was filtered using a Gaussian filter ($\sigma$=2). The filtered image was then segmented using a saturation threshold of 50% (inclusive). Object labeling was performed and objects smaller than 700 pixels were removed to focus on the upward moving front and to eliminate small regions of high

liquid water content which were more likely to be artifacts. The maximum height of all pixels in the segmented image was then used as the wetting front position. This height was added to the pressure head (in cm) at the bottom of the snow to provide the position with respect to the water table.

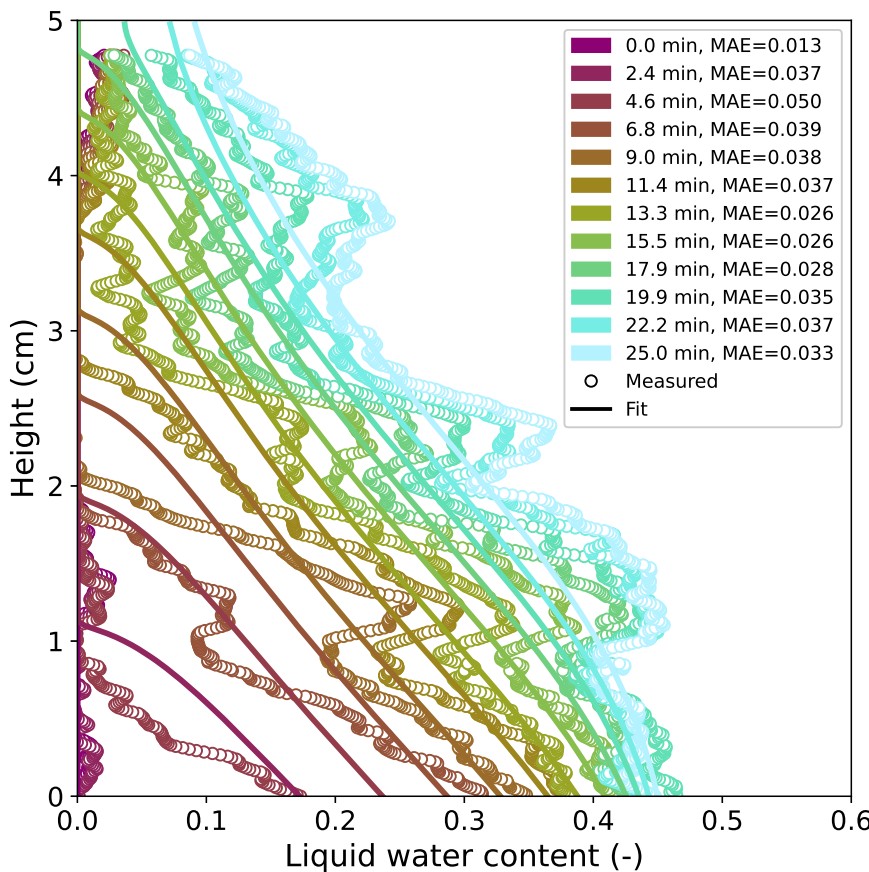

**Figure 5.** The temporal evolution of the vertical liquid water content profile (circles) of $FG_{s,g}$ and the accompanying results of the inverse fitting (lines) with mean absolute error (MAE) expressed as the volume fraction of liquid water content.

## 3  Results

Each of the 4 capillary rise experiments resulted in a time series of 2D optical density images, which were used to obtain 1D
and 2D information about the capillary rise process. An overview of $FG_{s,g}$ is provided in Fig. 6 and the corresponding overview plots for the other experiments are in the Appendix (Fig. A2–Fig. A4). The overview figures show the relationship between the 2D evolution of the optical density over time and the 1D profiles (optical density and liquid water content). Specifically, regions of higher optical density (vertically) in the 2D images correspond to regions with more water (and/or ice) and these regions have increased optical density and liquid water content in the 1D profiles.

When analyzing all 4 experiments, a few general results become clear. First, the 2D images show that the capillary rise process was not spatially homogeneous (Fig. 7). Second, the fine-grained snow led to higher capillary rise heights compared to the coarse-grained snow. Finally, the 1D optical density profiles show that the rate of capillary rise was higher for the

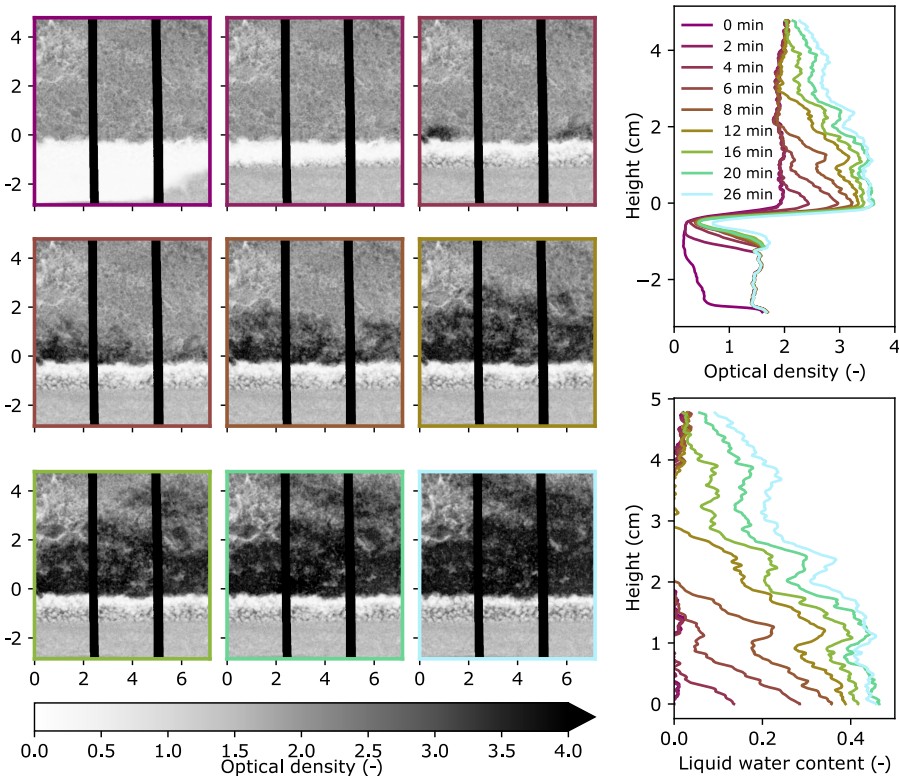

**Figure 6.** Summary figure of experiment FG$_{s,g}$ showing the 2D optical density evolution over time (left), 1D horizontally averaged optical density profile evolution over time (top right), and 1D horizontally averaged liquid water content profile of the snow over time (bottom right). The 2D images correspond to the times in the vertical profiles (right) as indicated by the color. The units for the optical density images (left) are cm for both axes. For the y-axes of all plots in this figure, a height of 0 cm was defined as the height where the continuous snow phase begins, where the effects of the gravel-snow interface are no longer present. The vertical black-body bars are shown in black in the images (left).

experiments without the porous gravel layer (Fig. 8). Thus, the properties of both the snow and the transitional layer below the snow influenced the capillary rise process.

## 3.1 Spatio-temporal variability

The 2D spatio-temporal variability of the capillary rise process was assessed by tracking the wetting front. An example of the wetting front progression is provided for FG$_{s,g}$ in Fig. 9, which shows the spatial and temporal inhomogeneity of the capillary rise process. The image is separated into three regions based on the black bodies. For FG$_{s,g}$, the water first rose in the left region (labeled as 1 in Fig. 9), but to a lower final height compared to the middle and right regions. In addition to the upwards flow direction, the figure also shows lateral flow patterns. For example, water flowed from the right region into the middle region

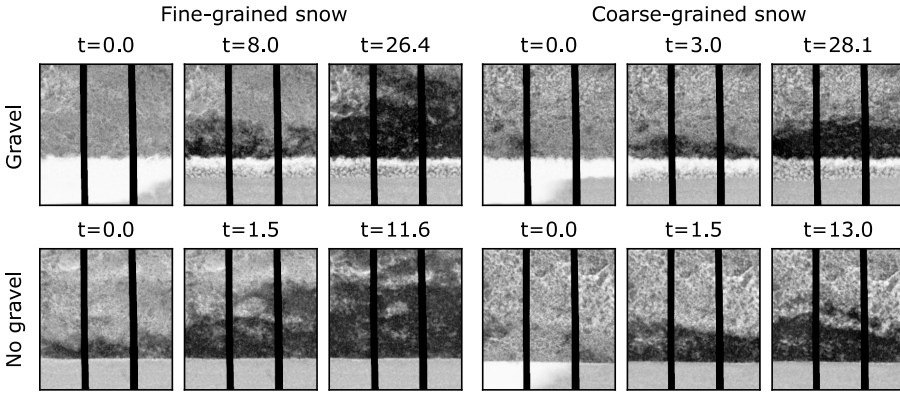

**Figure 7.** A summary of the 4 experiments showing the 2D optical density images at 3 times (t) given in minutes. The experiments are classified based on their snow type and whether or not they included a gravel layer. These correspond to $FG_{s,g}$ (top left), $FG_s$ (bottom left), $CG_{s,g}$ (top right), and $CG_s$ (bottom right). The vertical black-body bars are shown in black.

from 15 min to 20 min, between 3 cm and 4 cm (labeled as 2 in Fig. 9). At the end of the experiment, the three regions showed different wetting front patterns.

Some of the spatio-temporal variability is likely related to variations in snow properties. For example, regions of $FG_{s,g}$ that remained less saturated were characterized by particularly low densities with larger pores and weaker capillary forces (Fig. 10). Additionally, the fine-grained snow led to higher final wetting front positions (7 cm to 8 cm) compared to the coarse-grained snow (4 cm to 5 cm) as seen in Fig. 8 and Fig. 11. It should be noted that the wetting fronts in both experiments with the fine-grained snow ($FG_{s,g}$ and $FG_s$) reached the top of the image, so the maximum capillary rise heights are unknown. There were also differences in the final wetting front heights for each of the three image regions, on the order of 0.5 to 1.5 cm. Differences in the wetting front position during the experiments were larger, up to several centimeters. The wetting fronts tended to progress in sudden, relatively large increases as opposed to a smooth progression. Overall, $FG_{s,g}$ took the longest to reach its final wetting front position ($\sim$15 min–20 min depending on the region), while the other experiments reached their final positions in approximately 5 min. The wetting front tracking also highlighted the presence of water in the initial image (e.g., black circles in $CG_{s,g}$ in Fig. 11).

## 3.2 Flow rates

While the snow properties affected the capillary rise height, the properties of the transition layer under the snow affected the flow rates (Fig. 12). The experiments without the gravel layer ($FG_s$ and $CG_s$) exhibited higher maximum flow rates compared to the experiments with the gravel layer ($FG_{s,g}$ and $CG_{s,g}$). These calculated flow rates are roughly an order of magnitude lower than the saturated hydraulic conductivities determined via the inverse fitting (described in detail below), which is related to the unsaturated conditions in the transition layer. Additionally, the flow rates tended to drop to zero a few minutes after the wetting fronts reached their final position. For $FG_{s,g}$, the flow rates dropped to zero after 20 min, while this drop occurred between 5

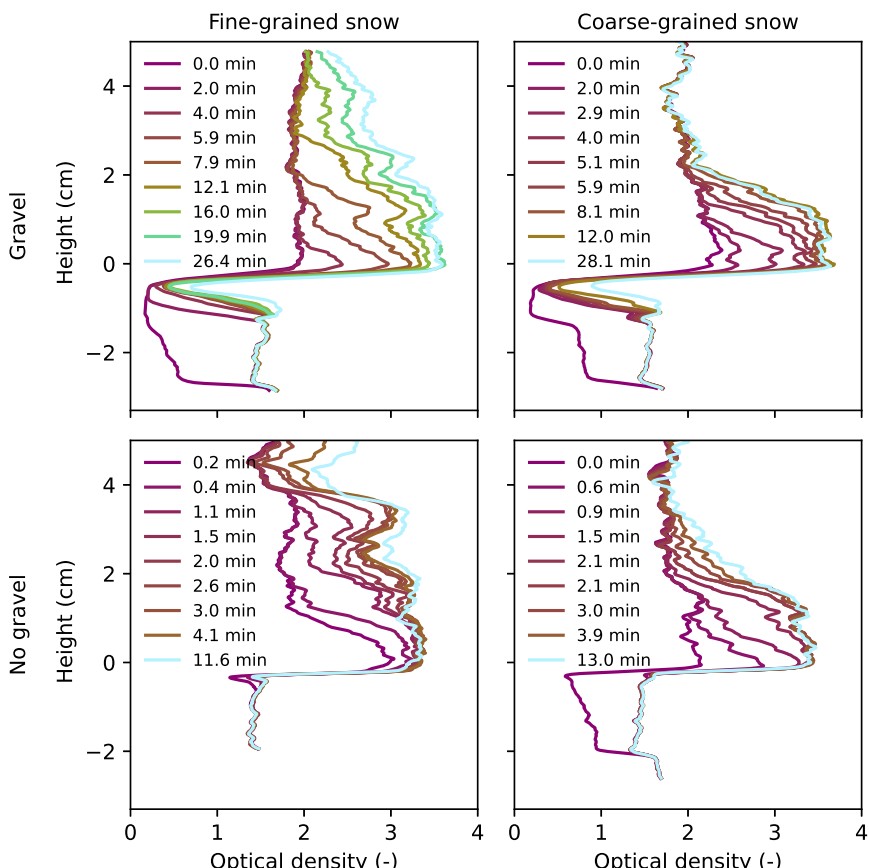

**Figure 8.** A summary of the 4 experiments showing the 1D optical density profiles at 9 times. The experiments are shown based on their snow type and whether or not they included a gravel layer. These correspond to FG$_{s,g}$ (top left), FG$_s$ (bottom left), CG$_{s,g}$ (top right), and CC$_s$ (bottom right). The unsaturated gravel layer can be seen in the upper plots as the reduction in optical density below a height of 0 cm.

min and 10 min for the other three experiments. This demonstrates that regions behind the wetting front continue to gain water after the front has passed.

### 3.3 Fitting hydraulic parameter values

The results of the fitting of the water retention curve (Fig. 4, Fig. A5–Fig. A7) and inverse fitting of temporal dynamics (Fig. 5, Fig. A8–Fig. A10) are provided in Tab. 1 with comparison to literature values using standard methods for determining these parameters. The MAE for the static fits ranged from 0.019–0.041 liquid water content (Fig. 4, Fig. A5–Fig. A7). The MAE for the inverse fits ranged from 0.013–0.076 liquid water content and was calculated for each timestep (Fig. 5, Fig. A8–Fig. A10).

The fit saturated liquid water content values ($\theta_s$) varied less than 10% from those estimated based on the optical density and $\mu$CT scans. The $\alpha$ parameters for the experiments with the fine-grained snow (FG$_s$ and FG$_{s,g}$) resulted in similar values

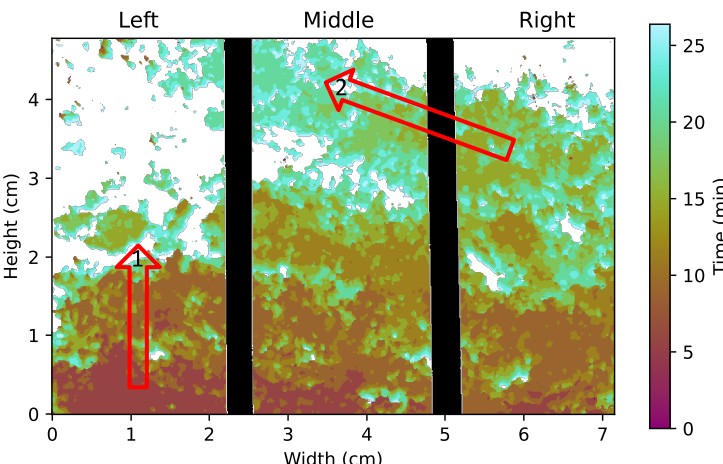

**Figure 9.** 2D wetting front evolution for fine-grained snow with gravel layer ($FG_{s,g}$). The times in the colorbar indicate when 50% saturation was achieved. White regions never reached 50% saturation. The black columns indicate the positions of the black-body bars. Key movements are labeled with red arrows (1) initial infiltration in the Left region and (2) horizontal flow from the Right to the Middle region

of 0.21 cm$^{-1}$ and 0.19 cm$^{-1}$, respectively. The $\alpha$ values for the experiments with the coarse-grained snow ($CG_s$ and $CG_{s,g}$) varied more at 0.22 cm$^{-1}$ and 0.30 cm$^{-1}$, respectively. The $n$ values followed the same pattern with lower and similar $n$ values of 4.5 and 3.5 for the experiments with fine-grained snow ($FG_s$ and $FG_{s,g}$), with larger and more dissimilar $n$ values of 18.6 and 7.3 for experiments with the coarse-grained snow ($FG_s$ and $FG_{s,g}$), respectively. The van Genuchten parameters resulting from the water retention curve fitting significantly differed from the values obtained with the parameterization by Yamaguchi et al. (2012). The $\alpha$ values obtained by the fitting were approximately 2–4 times larger than parameterized values when using the grain diameter visually estimated using a crystal card grid. Conversely, the $n$ values obtained via the parameterization by Yamaguchi et al. (2012) were 1-4 times larger than those obtained by fitting. The smaller $n$ values obtained via fitting correspond to a wider pore size distribution compared to the parameterization.

Similarly, the saturated hydraulic conductivity ($K_s$) values for the snow predicted by the parameterization of Calonne et al. (2012) were 1-6 times larger than the values obtained by the inverse fitting. There was no clear difference between the fine-grained and coarse-grained snow. The fitting resulted in $K_s$ values for the transitional gravel layer of 0.33 cm min$^{-1}$ and 3.41 cm min$^{-1}$ for $FG_{s,g}$ and $CG_{s,g}$, respectively. Fitting of $FG_s$ and $CG_s$ resulted in $K_s$ values for the transitional sand layer of 0.16 cm min$^{-1}$ and 0.15 cm min$^{-1}$, respectively. As discussed in the following section, these values reproduce the complexities of the hydraulic conductivity function of the transitional layer.

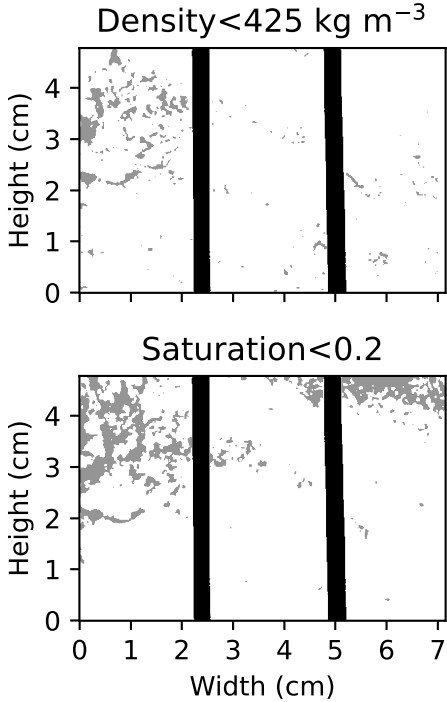

**Figure 10.** Segmented images of FG$_{s,g}$ showing regions with dry densities less than 425 kg m$^{-3}$ in gray (top image) and regions with final saturation values less than 20% in gray (bottom image). The black bodies are shown in black. The region of low saturation in the upper right of the saturation plot is an artifact of high initial density due to surface melting, which caused a high initial density and therefore a small increase in saturation.

## 4  Discussion

Neutron radiography was used to image the capillary rise of water from sand and gravel into snow. The images showed the 2D evolution of the liquid water distribution and fitting allowed for quantification of hydraulic parameter values of the van Genuchten model and saturated hydraulic conductivity. The parameter values were compared to literature values.

### 4.1  Flow rates

The flow rate of water into the snow was predominantly influenced by the transitional layer below the snow. As seen in Fig. 12, the presence of the unsaturated transitional gravel layer below the snow led to reduced flow rates compared to the configurations with only the sand. This is because the hydraulic conductivity of a material increases with saturation in a highly non-linear way (Eq. 12). Since the transitional gravel layers were relatively dry while the transitional sand layers were relatively wet, the

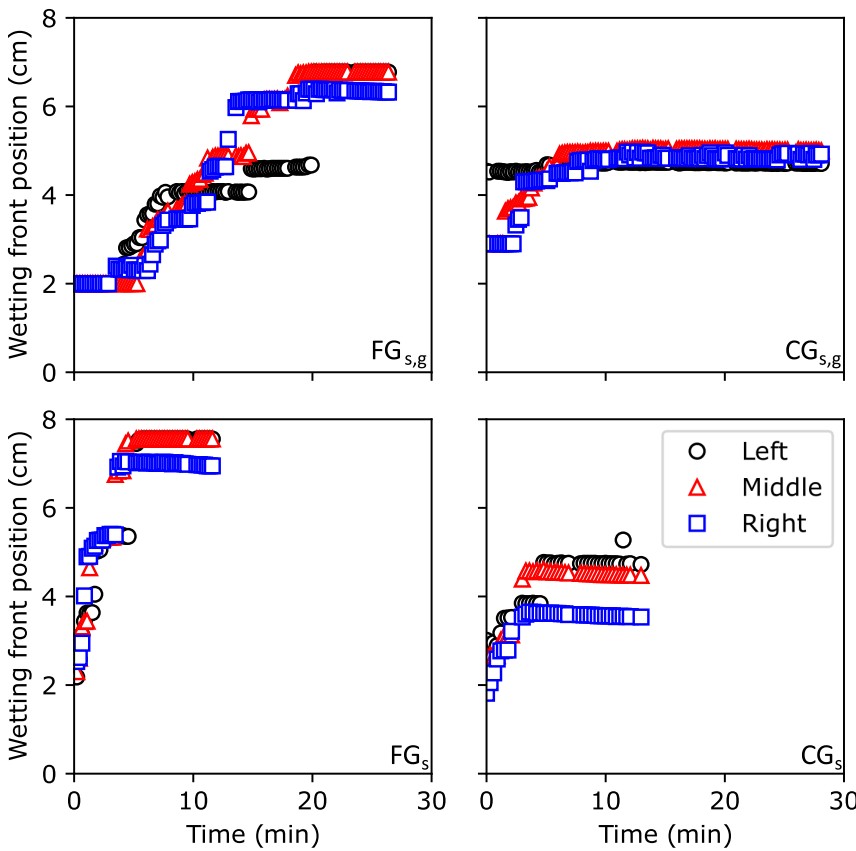

**Figure 11.** The progression of the wetting fronts for the three regions between each black-body for each experiment. The wetting front position is provided with respect to the water table. The legend in $CG_s$ applies to all plots in the figure.

hydraulic conductivity of the transitional gravel layer was lower. The flow rates did not show any distinction with respect to
335    the snow properties.

The flow rates were also significantly lower (1–2 orders of magnitude) than the saturated hydraulic conductivity values of the snow (inverse fitting and parameterized) and sand (Lehmann et al., 2008). This was likely due to a combination of factors. First, the sand and gravel were packed dry, which led to incomplete saturation. For example, the sand reached a saturation between 76% and 82% when using the apparent density provided by the manufacturer to calculate the dry porosity. No value could
340    be calculated for $FG_s$ because the sand was already wet in the first image. As mentioned above, since hydraulic conductivity increases with saturation, this resulted in lower flow rates than if the sand were perfectly saturated. Second, the imperfect gravel-snow and sand-snow interfaces were likely more porous than either the gravel or sand themselves, respectively, resulting in slower forms of flow such as corner and film flow compared to flow through saturated pore bodies. Finally, the flow was

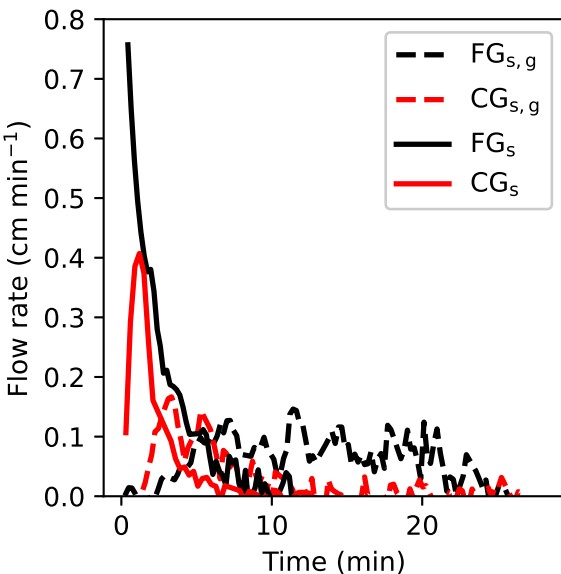

**Figure 12.** Flow rates of water into the snow for each experiment deduced from the images and calculated based on Eq. 9. The dashed lines represent the experiments with the interfacial gravel layer.

**Table 1.** Parameter values determined in this study and comparison values from the literature

| Experiment | Measured snow properties | | | | Fitted parameters[g] | | | | | Comparison values | | |
|---|---|---|---|---|---|---|---|---|---|---|---|---|
| | $\rho_{OD}$[a] | $\rho_{\mu CT}$[b] | $d_{\mu CT}$[c] | $d_{CC}$[d] | $\theta_r$ | $\theta_s$ | $\alpha$ | $n$ | $K_s$ | $\alpha$[e] | $n$[e] | $K_s$[f] |
| | kg m$^{-3}$ | kg m$^{-3}$ | mm | mm | – | – | cm$^{-1}$ | – | cm min$^{-1}$ | cm$^{-1}$ | – | cm min$^{-1}$ |
| FG$_{s,g}$ | 540 | 531 | 0.5 | 0.5 | 0.00 | 0.46 (0.005) | 0.21 (0.004) | 4.5 (0.3) | 3.3 (0.7) | 0.054 | 14.0 | 5.5 |
| FG$_s$ | 502 | 531 | 0.5 | 0.5 | 0.00 | 0.43 (0.01) | 0.19 (0.01) | 3.5 (0.5) | 2.9 (0.3) | 0.058 | 13.4 | 9.1 |
| CG$_{s,g}$ | 560 | 547 | 0.8 | 1.0 | 0.00 | 0.43 (0.003) | 0.22 (0.0005) | 18.6 (0.6) | 1.7 (0.4) | 0.102 | 9.7 | 10.9 |
| CG$_s$ | 487 | 547 | 0.8 | 1.0 | 0.00 | 0.46 (0.003) | 0.30 (0.001) | 7.3 (0.2) | 18.0 (3.9) | 0.118 | 9.0 | 28.2 |

[a] Density calculated from the neutron images

[b] Density calculated from the $\mu$CT scans

[c] Optically equivalent diameter based on the SSA from the $\mu$CT scans

[d] Average grain size using crystal card grid

[e] Based on Yamaguchi et al. (2012) using d$_{CC}$

[f] Based on Calonne et al. (2012) using d$_{\mu CT}$

[g] Standard error provided in parenthesis for $K_s$ and standard deviation for the static fit parameters

influenced by the pressure gradient. During infiltration, regions of the sample below the water level experienced a positive pressure head (upward force) while regions above the water level experienced a negative pressure head (downward force).

## 4.2 Fitted values - water retention curve fitting

The MAE of the water retention curve fitting varied from 0.019–0.041 liquid water content, with $FG_s$ having the highest MAE of 0.041 and the remaining experiments having similar MAEs of 0.019–0.023. The MAE can be used in combination with the standard deviation of the fit values (Tab. 1) to assess the overall quality of the fit as well as the confidence in the individual fit parameters. For example, the largest standard deviations for the $\alpha$ and $n$ parameters were obtained in the fit for $FG_s$, which also had the largest MAE. Additionally, for $FG_{s,g}$, $CG_s$, and $CG_{s,g}$, a standard deviation in the $\alpha$ parameter of up to 2% and a standard deviation in the $n$ of up to 7% led to an MAE of about 0.02. An MAE of 0.02 is similar to the errors of 2%–5% reported for high-resolution liquid water content measurement techniques such as calorimetry and MRI (Adachi et al., 2020), while Barella et al. (2024) reported a much higher accuracy of 0.5% for calorimetry. Thus, the fitting itself has an MAE that is similar to the error of the best measurement techniques and, considering the limitations and assumptions (e.g. 1D, homogeneous density) of the fits, indicates that the fit parameters are reasonable for the overall sample. The assumptions and limitations for the fitting procedure and their effect on the fit are described in more detail below (Section 4.4).

The van Genuchten parameter values $\alpha$ and $n$ from the water retention curve fitting were compared to the values based on the commonly used parameterization by Yamaguchi et al. (2012) (Tab. 1). The $\alpha$ values predicted by Yamaguchi et al. (2012) (comparison values in Tab. 1) are between 2 and 4 times smaller than the values obtained by fitting (fitted parameters in Tab. 1). Part of this discrepancy is likely due to hysteresis. Hysteresis is responsible for the offset between the wetting and drying curves of a water retention curve. For snow, the hysteresis factor for $\alpha$ between the wetting and drying curves has been experimentally measured at 1.5 (Adachi et al., 2020) and estimated through modeling to be >2.0 (Leroux et al., 2020). Contrarily, Coléou et al. (1999) found similar heights of the capillary fringe for wetting and drying of pre-wetted snow samples. For soils, a factor of 2 is often used (Kool and Parker, 1987). The ratios obtained here are on the high end of this range and may indicate that other factors increased this apparent hysteresis.

One of these factors is the discrepancy between the methods used to measure the grain size in the parameterization by Yamaguchi et al. (2012) (camera) and in this study (crystal grid card). For example, reducing the grain size of the fine-grained snow from 0.5 mm to 0.25 mm results in an $\alpha$ ratio close to 2. Variations of 0.25 mm in the grain size are well within the measurement error of the visual approach using the crystal grid and the method itself is different from the technique used by Yamaguchi et al. (2012). This discrepancy in grain size measurement methods is a known issue and the results presented here further underline the need for consistency between measurement techniques (Lombardo et al., 2025b). Therefore, it is unclear if the apparent hysteresis in $\alpha$ is really greater for the fine-grained snow compared to the large-grained snow.

For $n$, the values obtained by the parameterization by Yamaguchi et al. (2012) were 3 and 4 times larger than those obtained by the fitting for $FG_{s,g}$ and $FG_s$, respectively. For $CG_{s,g}$ and $CG_s$, the ratios were 0.5 and 1.2, respectively. These discrepancies are more difficult to understand. The relatively large ratios for the fine-grained snow experiments could be a result of the fact that the parametrization by Yamaguchi et al. (2012) was based on measurements with sieved snow, while the snow samples

used here were not sieved. Sieving leads to narrower grain size, and thus also pore size, distributions, which are characterized by higher $n$ values (van Lier and Pinheiro, 2018). Hysteresis in $n$ is typically not considered for snow since the experiments by Adachi et al. (2020) showed that it was minimal. However, the experiments by Adachi et al. (2020) consisted of only three measurements using sieved snow with grain sizes greater than 1 mm. Since the fine-grained snow used here was less than 1 mm in diameter, it is possible that there is a grain-size dependence on the hysteresis of snow. However, since we cannot differentiate between possible hysteresis and the effects of sieving, additional studies are needed to determine if hysteresis in $n$ exists for snow.

The $n$ values of the coarse-grained snow experiments are inconsistent due to the high $n$ value of $CG_{s,g}$ (18.6). This value is even higher than that for relatively monodisperse sands (Benson et al., 2014). Therefore, it seems this value is inflated. This was likely caused by a change in density at a height similar to the capillary rise height (Fig. A3), since a reduction in density would lead to a reduction in liquid water content. The true value is therefore likely lower. Since the $n$ value for $CG_s$ is relatively close to the predicted value by Yamaguchi et al. (2012), it seems reasonable to conclude that the coarse-grained snow showed better agreement in $n$ with the literature values than the fine-grained snow.

Differences in the van Genuchten parameter values between this study and literature values can also be related to the spatial variability caused by the asymmetric sample geometry. In contrast to the circular sample cross sections with diameters of 5, 8, and 15 cm in Yamaguchi et al. (2012) and Adachi et al. (2020), we used a rectangular cross-section with one dimension limited by the neutron beam attenuation (1 cm in the beam direction). The filling of this narrow shape is expected to be more heterogeneous compared to the sieving into larger, cylindrical columns used by Yamaguchi et al. (2012) and Adachi et al. (2020). This is manifested in density values that are considerably lower in some regions compared to the averaged dry density (see regions in Fig. 10 with density values below 425 kg m$^{-3}$ compared to averaged dry density of about 530 kg m$^{-3}$). Such spatial variability is directly related to the expanded width of the pore size distribution as expressed by smaller values of shape parameter $n$. In addition, the presence of regions with larger porosities, and thus larger pore sizes, aligns with the experimental findings of large $\alpha$ values. This can be seen, for example, in Tab. 1 for $FG_{s,g}$ with an averaged porosity of 0.41, which is smaller than the saturated water content of 0.46. The capillary forces in larger pores are generally weaker and the $\alpha$ value is correspondingly larger compared to a denser packing.

The use of the dry density calculated from the measured optical density ($\rho_{OD}$ in Tab. 1) as opposed to the dry density measured prior to packing ($\rho_{\mu CT}$ in Tab. 1) also had an effect on the water retention curve fitting. Generally speaking, the use of a different density shifts the water retention curve, leading to new fit values. For example, when comparing the fit parameters for $FG_{s,g}$ calculated with $\rho_{OD}$ compared to $\rho_{\mu CT}$, the dry density decreased by 2%, which led to an increase in the saturated liquid water content ($\theta_s$) of 2% (assuming $\theta_s$ is equal to the porosity), a decrease in $n$ of 3%, and a decrease in $\alpha$ of 0.5%. Changes in density do not affect the residual liquid water content ($\theta_r$) of these fits since the value was limited to near zero. The results were similar for $FG_s$, where $\rho_{\mu CT}$ was 6% larger than $\rho_{OD}$, which led to a decrease in the saturated liquid water content ($\theta_s$) of 8%, an increase in $n$ of 9%, and an increase in $\alpha$ of 2%. Thus, changes in dry density predominantly impact the saturated liquid water content and $n$ parameters. The relatively smaller change in $\alpha$ makes sense since the change in density

stretches or shrinks the curve horizontally (with respect to liquid water content), but not vertically (with respect to pressure head).

## 4.3 Fitted values - hydraulic conductivity fitting

Similar to the water retention curve fitting, the MAE can also be used for describing the quality of the inverse fitting of the temporal dynamics. The MAE of the inverse fitting varied from 0.013–0.076 liquid water content, with the MAE generally improving (i.e. becoming smaller) with time. In other words, the fit was generally better towards the end of the experiment, which was due to discrepancies between the simulation and measurements at the beginning of the experiments (discussed in detail in Section 4.4 below). It should be noted that the small MAE values in the first time steps are caused by the thick layer

of residual water content; the MAE within the wetted region is considerably larger. The inverse fitting was therefore poorer than the water retention curve fitting. This is further supported by the observation that the experiment (FG$_s$) with the smallest standard error in the saturated hydraulic conductivity had the largest MAE for the water retention curve. Thus, there is less certainty in the saturated hydraulic conductivity values ($K_s$) compared to the van Genuchten parameters obtained by the water retention curve fitting. This is likely due to additional fitting of the interfacial layer, which added another free parameter as well

as the general increased complexity of the inverse fitting (multiple layers and time dependence) compared to the static water retention curve fitting. These complexities are discussed in more detail below and in Section 4.4.

The saturated hydraulic conductivity values for the snow obtained via the inverse fitting in this study were 20%–60% of the values obtained by the parameterization by Calonne et al. (2012) and are generally consistent with measured the values by Katsushima et al. (2013). A direct comparison with the values by Katsushima et al. (2013) is not possible since SSA was not

measured in that study, and the density and grain sizes varied from those used here. In general, however, underestimation of $K_s$ with respect to the parametrization of Calonne et al. (2012) is expected. As described above for the sand, 100% saturation is difficult to achieve experimentally (Yamaguchi et al., 2010) and rapid infiltration of a dry sample traps air within the pores leading to a reduction in saturation. Both of these processes reduce the hydraulic conductivity (Faybishenko, 1995). The trapping of air during infiltration is not currently accounted for in snowpack models and leads to an overestimation of infiltration

rates at short time scales.

The $K_s$ value of the transitional layer below the snow was also a free parameter in the inverse fitting. For the experiments without the gravel layer (FG$_s$ and CG$_s$), the inverse fitting resulted in $K_s$ values for the transitional sand layer of 0.16 cm min$^{-1}$ and 0.15 cm min$^{-1}$, respectively. These values are 100 times smaller than the reported $K_s$ value of 15.6 cm min$^{-1}$ for this sand (Lehmann et al., 2008). As described above for the flow rates, this is likely due to a combination of factors including

dry packing and the braking effect of the gravel-snow and sand-snow interfaces. Additionally, while $K_s$ was the only free parameter in the fitting, it is not the only relevant parameter. Here, the $\tau$ parameter, related to the connectivity and tortuosity of the material, was fixed to the default value of 0.5 (Eq. 12). The model was therefore forced to capture changes in $\tau$ with $K_s$. The inclusion of the $\tau$ parameter has been suggested as a way to improve estimations of hydraulic conductivity of snow (Yamaguchi et al., 2010). Thus, it is not surprising that the $K_s$ values for the transitional sand layer differ from the reported

value.

These effects are also relevant for the experiments with the transitional gravel layer (FG$_{s,g}$ and CG$_{s,g}$), which resulted in $K_s$ values of the transitional gravel layer of 0.3 cm min$^{-1}$ and 3.4 cm min$^{-1}$, respectively. As with the sand layer, it is to be expected that the dry gravel packing and unsaturated gravel-snow interface led to a relatively small effective hydraulic conductivity. Compared to the sand, the gravel should have a larger saturated hydraulic conductivity as it has larger pores. As such, the

fact that these values are larger than those for the transitional sand layer is reasonable. However, it is unclear exactly why the $K_s$ values for the transitional gravel layer differ by an order of magnitude between FG$_{s,g}$ and CG$_{s,g}$. The discrepancy may be due to a poor definition of the hydraulic properties of the transitional layer. The conductivity values at the interface are highly dependent on the liquid configuration within the large interfacial pores. For such a thin region consisting of only a few pores, the formulation of representative hydraulic properties is difficult, considering that the conductivity values are controlled by

only these few interfacial pores.

## 4.4    Limitations of the fitting

Both the fitting of the water retention curve and inverse fitting of the temporal dynamics were fundamentally limited by the uncertainties in determining liquid water content (described in detail below). Thus, the accuracy of the values are proportional to the quality of the liquid water content (more specifically the dry snow density) estimations. The associated errors could be

reduced with better execution of the experimental procedure related to inconsistent packing, temperature control, and the delay prior to the first image.

The fitting processes themselves also have some intrinsic limitations. For example, both fitting methods were 1D and used a single set of van Genuchten parameter values for the snowpack and transitional layer below the snow. This means that lateral variations in properties were smoothed and the vertical inhomogeneities (e.g. snow density gradients) within each material

were not accounted for. These inhomogeneities are clearly seen in, for example, the strong dip in liquid water content at about 4.5 cm for FG$_s$ (Fig. A5 and Fig. A8) and the optical density images (Fig. 6 and Fig. A2–Fig. A4). Fitting with more materials and/or in 2D is possible and could improve results. However, the number of free parameters increases with the number of unique material layers and thus increases the complexity of the fitting. This is likely only beneficial with improved control of the snowpack properties and packing. A similar argument can be made for allowing the $\tau$ parameter to vary.

For the inverse fitting, there was a discrepancy between the setup of the initial conditions of the model and the physical system. The sand and gravel layers were initialized at the residual water content ($\theta_r$). However, as clearly seen in the initial images (Fig. 7), water had frequently already infiltrated a portion of the sand by the time the first image was taken. Thus, the first image in the experiments was often effectively ahead of the model, since the model was initialized completely dry at t=0. Interestingly, the models quickly overtook the experiments, resulting in higher liquid water contents compared to the

experiments even after only 1 timestep. This offset of initial conditions, in combination with the relatively small timesteps used for the fitting at the beginning of the experiment, likely contributed to errors in the inverse fitting (see discussion of MAE above).

There are also some limitations related to the use of MAE as in indicator for fit quality. This is because MAE weights each measurements equally. This has the consequence of treating the bottom of the sample (where most of the changes in liquid

water content occur) and the top of the sample (where the samples are mostly dry throughout the experiment) the same. Since the simulation is initialized with dry snow, the errors in liquid water content in the lower part of the snow as the water infiltrates are balanced by the many data points with a liquid water content of 0 in the upper part of the snow. This results in a lower MAE than may seem appropriate given the large discrepancy at the bottom of the snow. This effect is more pronounced for the inverse fitting since the water retention curve has a more balanced distribution of wet and dry measurements. This adds

additional uncertainty to the interpretation of the hydraulic conductivity fitting. Given the uncertainties of the experiments, analyses of future, more controlled experiments may benefit from an MAE weighted by the liquid water content or a different indicator of the fit quality (e.g. Wasserstein distance).

## 4.5 Limitations of neutron radiography

As mentioned above, the key limitation of these experiments is the inability to directly distinguish between optical density

contributions from water and ice. For the 1D liquid water content profiles, a single value for the dry density was applied to the entire sample (Fig. 8). For the 2D analyses (Fig. 7), a dry density was determined for each pixel. For both methods, the density was assumed to be constant throughout the course of the experiment. Several assumptions and limitations are associated with these approaches.

For both methods, changes in density over time were not considered. This is potentially problematic due to three processes:

melting, settling, and wet-snow metamorphism. Melting was mitigated by cooling the packed columns in an ice bath, using ice water in the reservoir, and through the use of the climatic chamber. However, no independent measurements of the sample temperature or infiltrating water were possible. Melting is assumed to have caused several artifacts such as some of the liquid water present in the initial images of $CG_s$ and $CG_{s,g}$, as well as the region of high density in Fig. 10. It may also have played a role in the movement seen within the snowpack of $FG_{s,g}$ as shown in the videos provided in the Video Supplement (Lombardo

et al., 2025a). Snow is also known to undergo metamorphism in the presence of water (Colbeck, 1986; Brun, 1989). Since the snow samples used in these experiments were melt forms, with the fine-grained snow also containing some small rounded grains, grain-coarsening was likely the predominant metamorphism process. Since grain-coarsening occurs on the timescales of hours (Colbeck, 1986), we do not expect wet-snow metamorphism to dramatically change the snow matrix over the duration of these experiments (maximum of 26 minutes). Settling of the snowpack is not expected to have played a role given the short

timescales and minimal snow heights (Marshall et al., 1999).

In addition to possible changes in the snow matrix over time, the initial density distribution was not homogeneous. This is seen in the 2D images (Fig. 7) as well as the 1D optical density profiles (Fig. 8). For the 1D profiles, the use of a single value ignored vertical density gradients, which were particularly prevalent in the coarse-grained snow experiments ($CG_s$ and $CG_{s,g}$). However, this method allowed for the correction of the initial dry density when water was present in the first image. Since the

dry density values for the 1D analyses were reasonable compared to the pre-packed values, errors in the 1D profiles are mostly due to the spatial distribution of the snow and likely had the largest effects on the water retention curve fitting.

For the pixel method used for the wetting front analyses, 2D variations in density were accounted for. However, the influence of water in the initial images was not. Thus, uncertainties in the wetting front position are likely for regions with high liquid water contents in the initial images as these dry densities were artificially high.

Ultimately, while neutron radiography provided a high spatio-temporal resolution, the large uncertainties required a reduction in resolution for the fitting. The method therefore provided results on a scale somewhere in between the larger scales typical in the field (>cm, e.g. Techel and Pielmeier (2011)) and the high resolutions of laboratory techniques such as micro-computed tomography and MRI (Adachi et al., 2020; Yamaguchi et al., 2025). Improvements to the experimental procedures, as outlined above, could reduce these uncertainties. Particularly helpful would be obtaining a truly dry image prior to water
infiltration. Another option would be to drain the water from the snow after the experiment by lowering the reservoir. The resulting snow structure with residual liquid water could help estimate changes to the snow structure. However, the use of other techniques that can directly differentiate between water and ice, such as time-of-flight radiography with cold neutrons or MRI, will be necessary for providing more detailed measurements (Siegwart et al., 2019; Yamaguchi et al., 2025).

## 4.6   Implications for future research

Our results have several implications for the future study of capillary-driven processes in snow and their implementation in snowpack models. First, the results suggest that wetting effects such as hysteresis in the water retention and hydraulic conductivity curves, and trapping of air within pores during infiltration should be implemented in snowpack models. Hysteresis has been measured previously and the results here further support the magnitude of the hysteresis (Adachi et al., 2020; Leroux et al., 2020). Additionally, the incomplete saturation of snow is typically assumed to be about 90% of the pore space (Yamaguchi
et al., 2010, 2012), but has been experimentally measured as low as 80% (Adachi et al., 2020). For top-down infiltration, this effect could be even larger as the air is trapped from above. This may have large effects on the dynamics of capillary-driven processes, which are currently not accounted for in most snowpack models.

    Second, at the soil-snow interface specifically, the presence of a porous vegetation layer was previously shown to have no effect on the liquid water content within the snow, given that the layers were hydraulically connected and in hydrostatic equi-
librium (Lombardo et al., 2025b). Here, under transient conditions, we showed that a relatively dry transitional layer below the snow slowed the flow of water into the snow substantially. Additionally, the timescales for the capillary rise experiments performed here were on the order of minutes, which should be considered when comparing to other relevant snowpack processes such as metamorphism. For glide-snow avalanches specifically, the results demonstrate the importance of considering the the hydraulic properties of the soil-snow interface.

Finally, the results highlight a constant challenge in snow science. Namely, that many discrepancies exist between measurement techniques and their spatio-temporal scales. For example, parameterizations are often determined with high-resolution techniques in the lab, while the parameterizations are then applied to field data with much lower resolution. This was described above for the parameterization by Yamaguchi et al. (2012) and is a good example of how mixing measurement techniques can lead to substantial errors and convolute results. Whereas for dry snow, micro-computed tomography has asserted itself as
the gold standard for microstructure measurements, no analogous technique has been proven for wet snow (Lombardo et al.,

2025c). Without better measurement techniques, wet snow research will continue to lag behind its dry counterpart. As climate change increases the amount of wet snow in alpine regions, the lack of reliable data on wet snow will become increasingly problematic for snow hydrology and avalanche forecasting models (Mayer et al., 2024).

## 5 Conclusions

Capillary rise experiments of sand-snow and sand-gravel-snow systems were performed using neutron radiography, providing high resolution (92 $\mu$m and 15 s), 2D images of the capillary rise process. The images allowed for quantification of snow water retention curves, flow rates, and hydraulic conductivities. The results showed that both snow grain size and density affected capillary rise height, while the properties of the transitional layer below the snow predominantly influenced the dynamics. When compared to literature, the comparatively large $\alpha$ parameter values obtained in these experiments support the existence

of hysteresis in the water retention curve of snow. Additionally, the experiments demonstrated that dry, porous interfaces (here, the gravel) can lead to large reductions in flow rates. The capillary rise process itself was relatively inhomogeneous at the mm to cm scale investigated here, which was shown to depend on the homogeneity of the snow properties. The limited amount of data on snow hydraulic properties and inconsistency between measurement techniques at the laboratory and field scale is problematic for the implementation of the available data in snow models, and will need to be resolved as climate change

increases the prevalence and importance of wet snow.

*Data availability.*  Data for the inverse fitting are available at https://doi.org/10.16904/envidat.572. Other data are available upon request.

*Video supplement.*  Videos of the experiments are available at https://doi.org/10.16904/envidat.572.

## Appendix A:  Appendix

### A1  Snow microstructure

The snow samples were imaged with $\mu$CT to obtain high-resolution measurements of the density and specific surface area (Fig. A1).

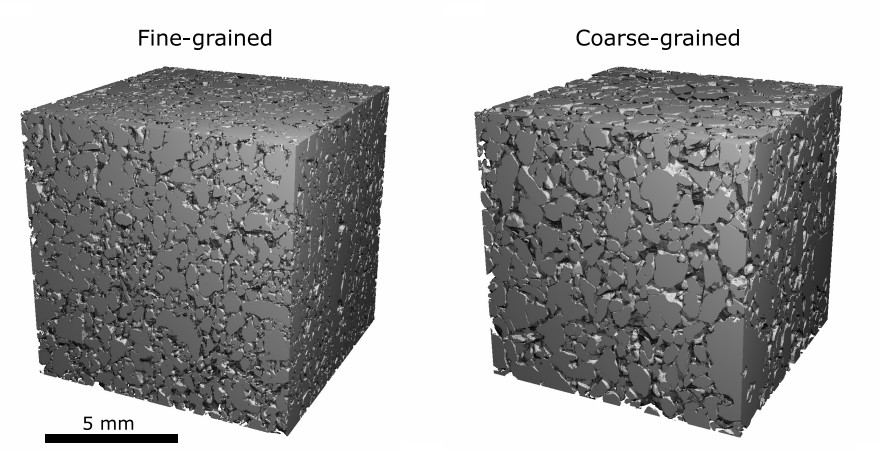

**Figure A1.** Micro-computed tomography images of the fine-grained (FG) and coarse-grained (CG) snow samples. The snow structure is shown in gray with the pore space in white. The images have a voxel size of 11 $\mu$m. Density and specific surface area are provided in Tab. 1.

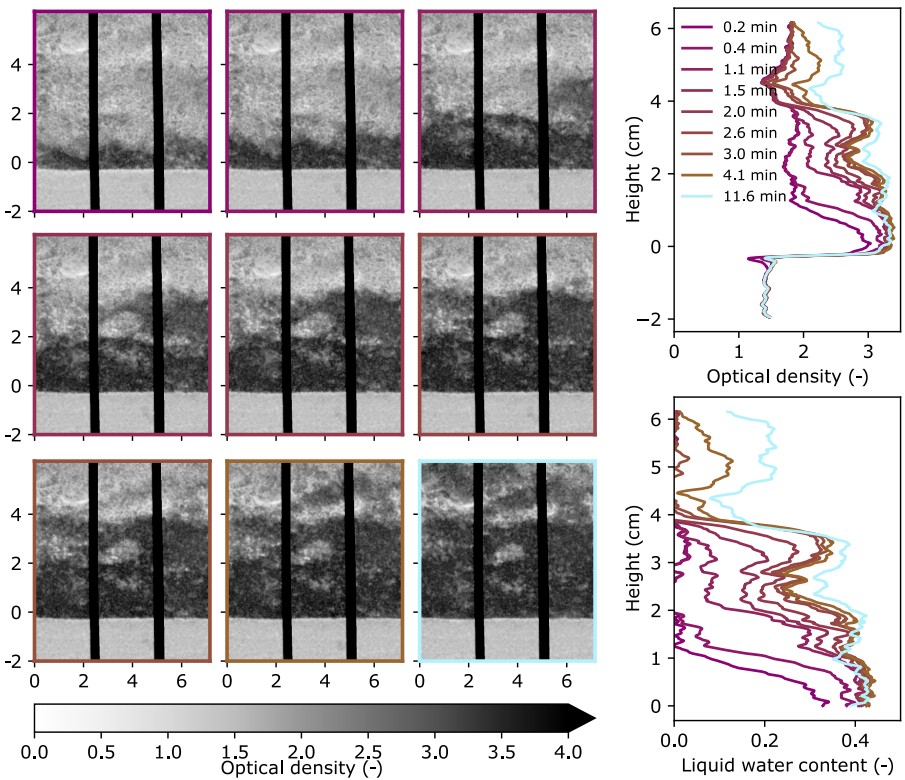

**Figure A2.** Summary figure of experiment FG$_s$ showing the 2D optical density evolution over time (left), 1D horizontally averaged optical density profile evolution over time (top right), and 1D horizontally averaged liquid water content profile of the snow over time (bottom right). The 2D images correspond to the times in the vertical profiles (right) as indicated by the color. The units for the optical density images (left) are cm for both axes. For the y-axes of all plots in this figure, a height of 0 cm was defined as the height where the continuous snow phase begins, where the effects of the sand-snow transition are no longer present. The vertical black-body bars are shown in black in the images (left).

## A2 Summary figures

The summary figures for FG$_s$, CG$_{s,g}$, and CG$_s$ show the 2D evolution of the optical density and the resulting 1D profiles (Fig. A2-A4).

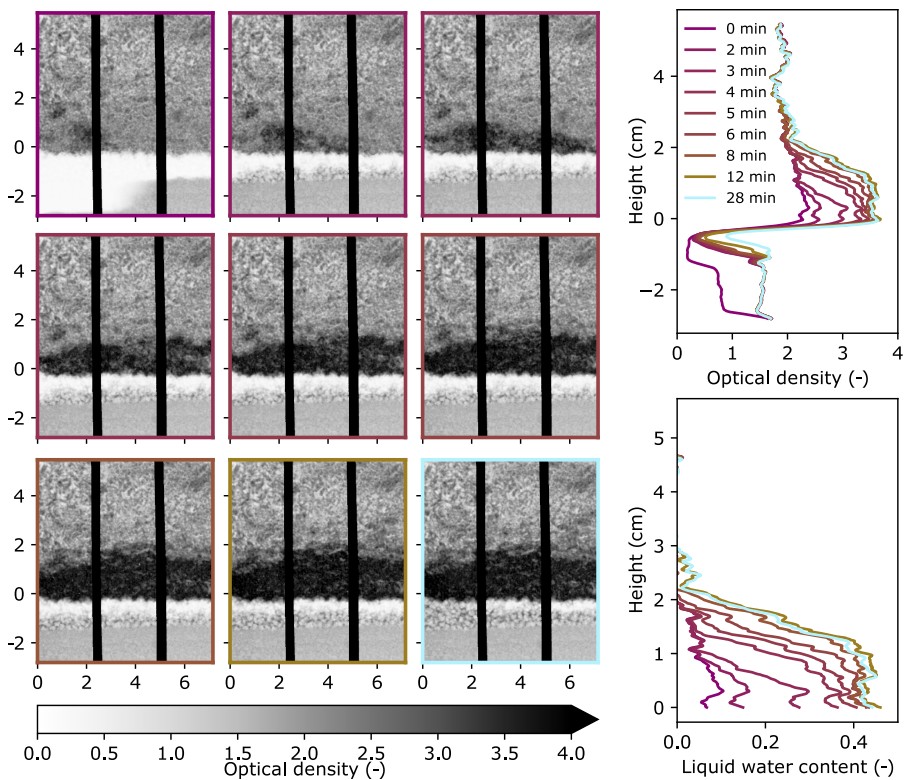

**Figure A3.** Summary figure of experiment CG$_{s,g}$ showing the 2D optical density evolution over time (left), 1D horizontally averaged optical density profile evolution over time (top right), and 1D horizontally averaged liquid water content profile of the snow over time (bottom right). The 2D images correspond to the times in the vertical profiles (right) as indicated by the color. The units for the optical density images (left) are cm for both axes. For the y-axis of all plots in this figure, a height of 0 cm was defined as the height where the continuous snow phase begins, where the effects of the gravel-snow interface are no longer present. The vertical black-body bars are shown in black in the images (left).

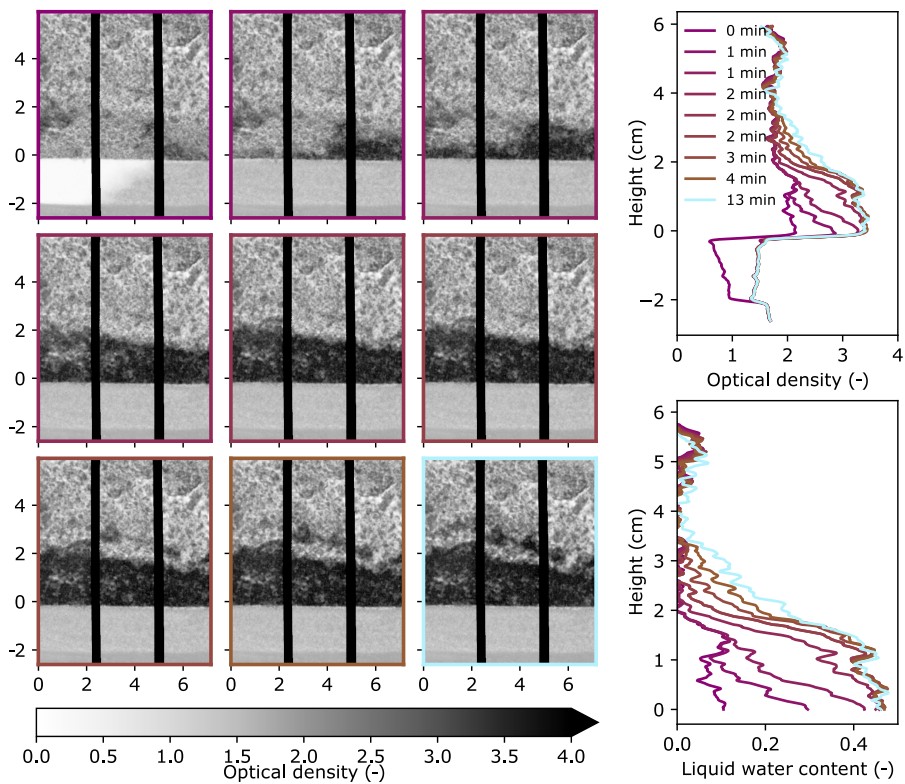

**Figure A4.** Summary figure of experiment CG$_s$ showing the 2D optical density evolution over time (left), 1D horizontally averaged optical density profile evolution over time (top right), and 1D horizontally averaged liquid water content profile of the snow over time (bottom right). The 2D images correspond to the times in the vertical profiles (right) as indicated by the color. The units for the optical density images (left) are cm for both axes. For the y-axis of all plots in this figure, a height of 0 cm was defined as the height where the continuous snow phase begins, where the effects of the sand-snow transition are no longer present. The vertical black-body bars are shown in black in the images (left).

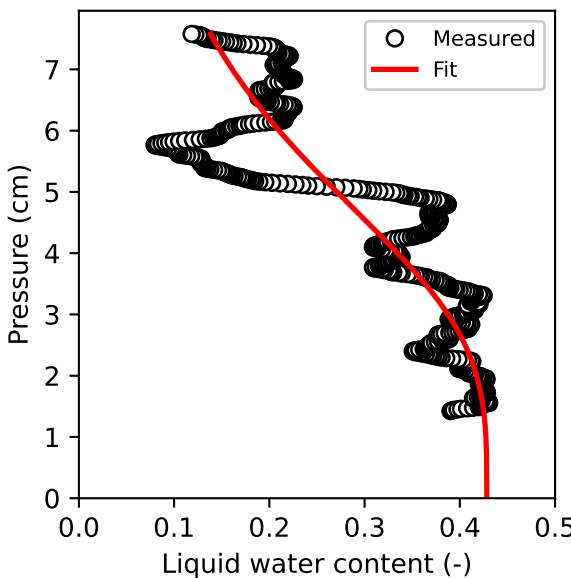

**Figure A5.** The 1D liquid water content of FG$_s$ as a function of pressure head (black circles) and the fit of these points to Eq. 10 (red line) with the mean absolute error (MAE) equal to 0.041 and expressed as the volume fraction of liquid water content.

**A3    Water retention curve fitting**

The water retention curve fitting for FG$_s$, CG$_{s,g}$, and CG$_s$ show measured liquid water content profiles and resulting fits (Fig. A5-A7).

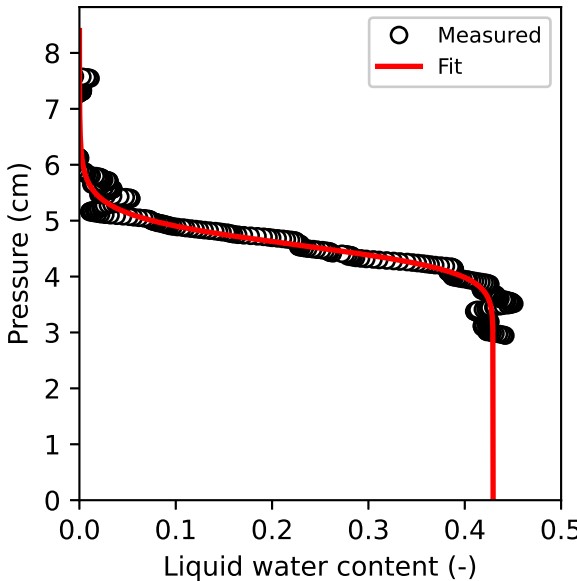

**Figure A6.** The 1D liquid water content of $CG_{s,g}$ as a function of pressure head (black circles) and the fit of these points to Eq. 10 (red line) with the mean absolute error (MAE) equal to 0.023 and expressed as the volume fraction of liquid water content.

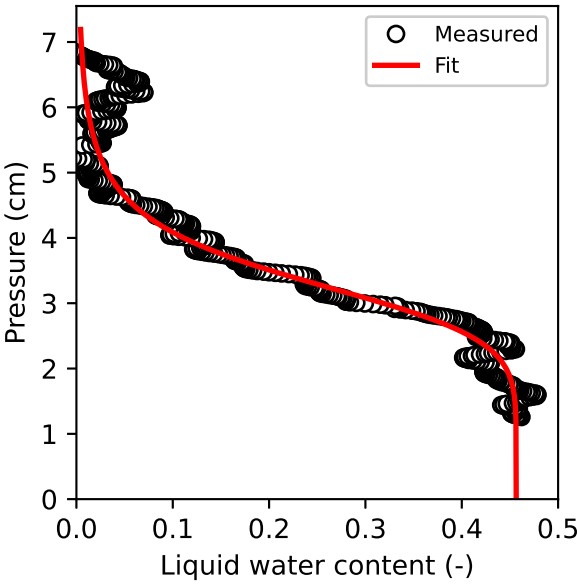

**Figure A7.** The 1D liquid water content of $CG_s$ as a function of pressure head (black circles) and the fit of these points to Eq. 10 (red line) with the mean absolute error (MAE) equal to 0.019 and expressed as the volume fraction of liquid water content.

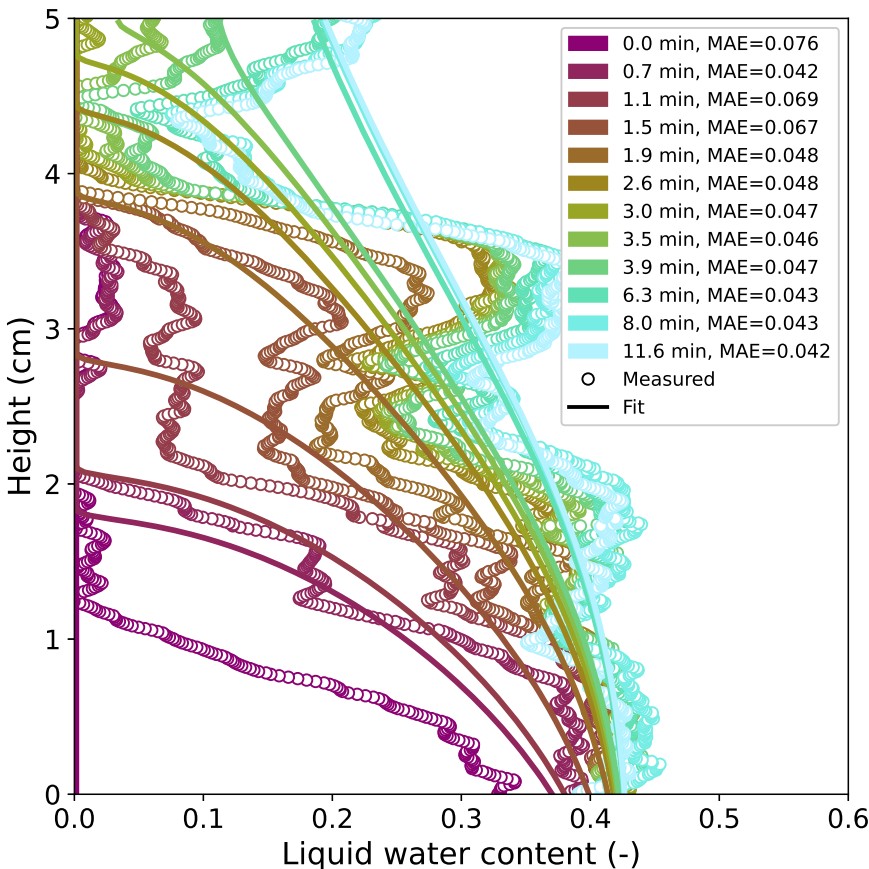

**Figure A8.** The measured liquid water content profiles and the accompanying inverse fit results for $FG_s$ with the mean absolute error (MAE) expressed as the volume fraction of liquid water content. Note that the small liquid water content values near 4.5 cm caused a large error in the fitting

## A4 Hydraulic conductivity fitting

The inverse fitting for $FG_s$, $CG_{s,g}$, and $CG_s$ show the evolution of the liquid water content profiles and resulting fits (Fig. A8-575  A10).

*Author contributions.* Conceptualization: All authors; Data curation: ML, AK; Formal analysis: ML, AK, PL; Funding acquisition: JS, PL; Investigation: ML, AK, AF; Methodology: ML, AK, AF, PL; Project administration: All authors; Resources: All authors; Software: ML, AK; Supervision: AH, JS, PL; Validation: ML, AK, PL; Visualization: ML; Writing - original draft preparation: ML; Writing - review and editing: All authors

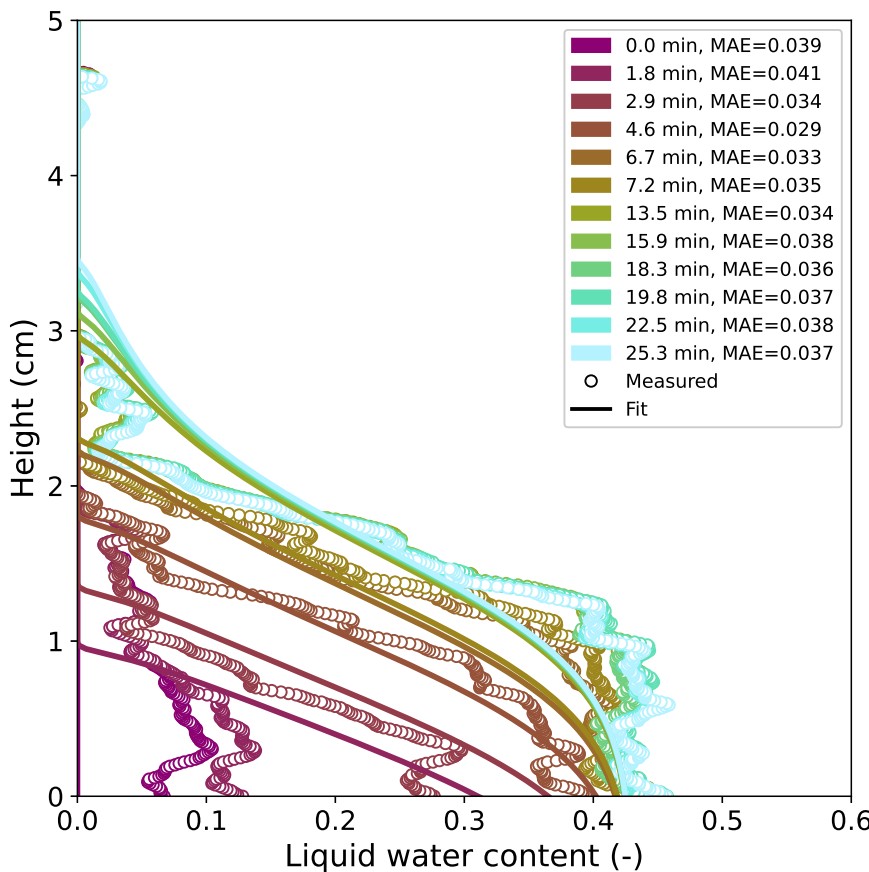

**Figure A9.** The measured liquid water content profiles and the accompanying inverse fit results for $CG_{s,g}$ with the mean absolute error (MAE) expressed as the volume fraction of liquid water content.

*Competing interests.* Jürg Schweizer is a member of the editorial board of The Cryosphere.

*Acknowledgements.* This work is based on experiments performed at the Swiss spallation neutron source SINQ, Paul Scherrer Institute, Villigen, Switzerland. This project was funded by SNSF Grant 200021-212949.

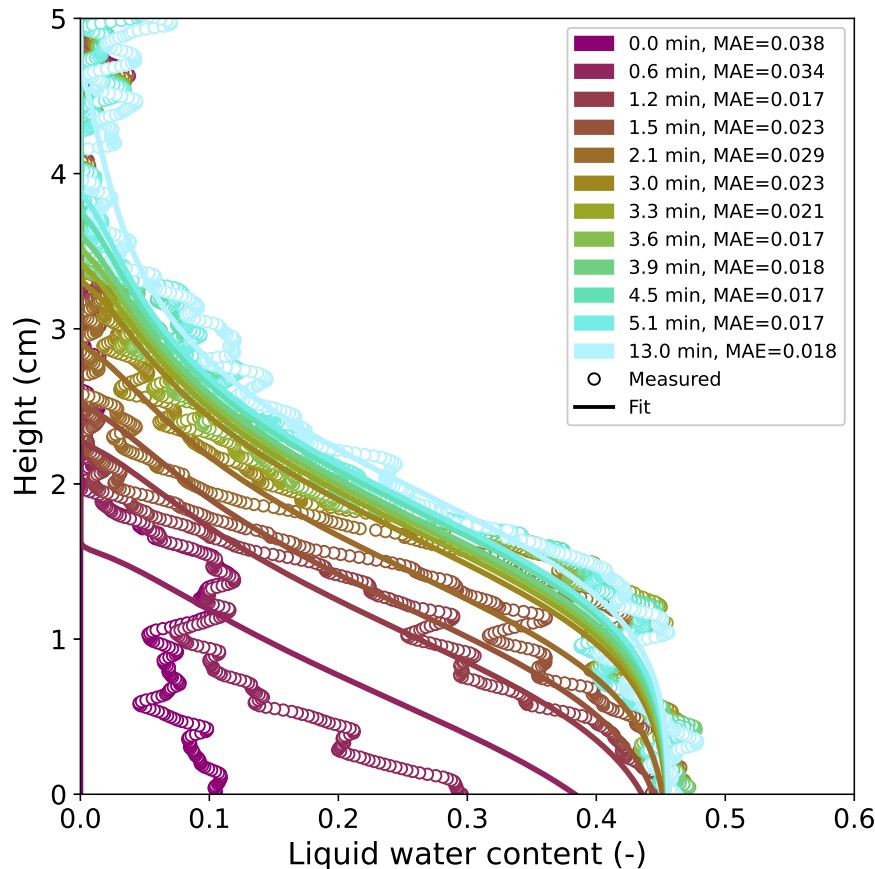

**Figure A10.** The measured liquid water content profiles and the accompanying inverse fit results for CG$_s$ with the mean absolute error (MAE) expressed as the volume fraction of liquid water content.

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
