# Peer review of "Quantification of capillary rise dynamics in snow using neutron radiography"

_EGUsphere, 2025_

## Author Response (AR1)

Reply to Reviewer 1

*We thank Reviewer 1 for their detailed review and suggestions for improving the manuscript. Each comment is addressed below with the reviewer's comment in normal black text and our response in italicized blue text.*

In this paper, the authors report on the results of measurements of capillary rise in snow, which is important not only for understanding the hydrological characteristics of snow but also for considering the interaction between snow and soil, using neutron radiography with high spatial and temporal resolution. In addition, based on the measurement results, they calculated the water retention curve and saturated hydraulic conductivity, which are important for understanding the characteristics of water movement in snow, and compared them with previous research.

The structure of the paper is very well organized and easy to understand. In particular, the experimental methods and analysis methods are described in detail, and this will be very useful when conducting additional experiments in the future. The results and discussion are highly reliable, as the study not only compares with previous research, but also takes into account the limitations of the measurement methods used in this study. The results of this research will undoubtedly contribute to the development of wet snow research, and are of sufficient scientific value to be published in academic journals.

The paper is of a high standard and there are no major points that need to be revised, but I will list some points that I noticed.

Major comments:
1. When determining the water retention curve and saturated hydraulic conductivity, the measured data is fitted using the least squares method. I understand that you have chosen the best method for determining the parameters, but in some cases, the fitting seems to be unreasonable depending on the experimental conditions. This is particularly true for the fitting used to determine the saturated hydraulic conductivity. I recommend that you show the regression error so that readers can judge the reliability of each fitting

   *The errors for each of the fitted parameters are included in Table 1 and we added some metrics for the quality of the fit ($R^2$) to the revised manuscript in Figures 4, A5-A7, A11 as well as accompanying text in Lines 426-432.*

2. The dry density of the snow sample is required to calculate the liquid water content, and they were calculated from their optical density of the first image. As Table 1 shows, the dry density calculated from the optical density differs even for the same snow quality (ex. $FG_{s, g}$, $FG_s$) and they also differ from the density obtained from X-ray CT. As the authors point out, it is not necessary for the density obtained from X-ray CT to match the density of the sample in the case. However, when calculating the liquid water content using the density obtained from X-ray CT, how much difference is there compared to when calculating the liquid water content using the density obtained from the optical density? Such information (The impact of density estimation on results) should be important for readers to understand the reliability of the results of this study and the points for improvement of this method. Therefore, I propose adding this kind of discussion to the paper.

   *We discuss this in more detail in the revised manuscript in Lines 368-377.*

Specific comments:
L81: How was the melt form created in an environment with -1 °C?

*Thank you for pointing this out – we adjusted the text to clarify this in Lines 85-87.*

L263: In the text, it is claimed that "the fine-grained snow led to higher final wetting front positions (7 cm to 8 cm) compared to the coarse-grained snow (4 cm to 5 cm)." Which diagram did you use to make this judgement?

*This can be seen in Figures 8 and 11 and we added this more explicitly to the revised manuscript (Lines 275-276).*

Reply to Reviewer 2

*We thank Reviewer 2 for their review. Each comment is addressed below with the reviewer's comment in normal black text and our response in italicized blue text.*

This paper presents neutron radiography results of snow/soil interface fitting it with a 1D model. The paper is very well written and clearly structured.

My main concern is heterogeneity: the sample is highly heterogeneous and so is the fluid flow (e.g., Fig. 9), yet the methods adopted essentially neglect this intrinsic heterogeneity.

*We do ignore the heterogeneity for the 1D profiles and quantification of the hydraulic properties, but extensively discuss these limitations in the manuscript. However, we do show aspects of this heterogeneity in Figures 9 to 11. We agree that 2D quantification is a necessary next step and believe this will be more beneficial in future experiments where issues such as premature melting are mitigated.*

The adoption of radiography (over neutron tomography) for example, could be questioned.

*In principle, tomography would be better. However, at NEUTRA, the flux is too low to perform tomography to allow us to capture the dynamics in these experiments. Additionally, regardless of the flux, the sample would have been limited to a column of about 1 cm in diameter, which would have limited the size of the sample dramatically. Tomography experiments at a higher flux source would be very interesting to perform.*

Even neglecting the third dimension, the fitting curves appear rather far from the experiments (e.g., Fig. A8/9/10) both in terms of local fluctuations and overall trend (this could also explain some of the convergence issues).

*We agree that there are deviations, particularly in the fits of hydraulic conductivity ($K_s$) and this is a necessary topic of future research. Based on suggestions by Reviewer 1 and the Editor, we added information about the quality of the fits to the manuscript (Lines 426-432, Figures 4, A5-A7, A11) to quantify this in addition to the errors in the fit parameters provided in Table 1.*

Much of the discussion is focused on the mismatch between the fitted parameters and literature predictions but this discussion could appear a bit stretched once accounting for this simplification.

*As discussed above, we are well aware of the limitations of the assumptions made here and discuss these in detail. However, we think that the comparison to literature is still valid (and important) given that the literature values also have their own limitations. For example, the van Genuchten parameters determined by Yamaguchi et al. (2012) were determined at a much lower spatial resolution (2 cm height, 5 cm diameter), so averaging across the entire sample seems appropriate for comparison. Similarly, the hydraulic conductivity experiments by Katsushima et al. (2013) also used larger samples. As such, we intend to leave the discussion as is and allow the reader to interpret the discussion given the limitations we outline throughout.*

In the 2D analysis you make the hypotheses that the snow does not move but playing the videos in the supplementary materials they all move by a significant amount downwards with a slight turn.

*This is indeed true. However, the analyses in Figures 9, 10, and 11 are meant to demonstrate that we can quantify 2D effects and to show that they are important to consider in the future. For the 1D analyses, the movement should not affect the results more than the assumption of a single density for the entire snowpack.*

Specific and minor notes:
= Section 2.4 appears scientifically correct and clear, but it follows mostly a well-established approach and could conceivably be moved to an Appendix.

*We think this section is important to provide an overview of the method for those less familiar with it.*

= The review in the introduction is paraphs a bit broad compared to the actual topic of the work and could conceivably be refocused. Also, it does not mention if a similar experimental approach has been adopted before.

*We are not sure what the reviewer means here exactly. We feel the introduction is very focused on describing water flow in snow and we provide an overview of the experiments and modelling approaches which have been used to date.*

= Why deduce the density from CT and not gravimetric measures?

*We chose to use the density obtained from the µCT scans because we used these measurements for the SSA. We also measured the density with a 100 $cm^3$ density cutter and it provided similar values.*

= "weak capillary forces of a high porosity layer of a vegetation layer" perhaps rephrase to avoid repetition?

*Good point – we rephrased this in Line 79.*

Reply to Editor

*We thank the Editor for their detailed review and suggestions for improving the manuscript. Each comment is addressed below with the reviewer's comment in normal black text and our response in italicized blue text.*

I consider that the manuscript meets the criteria to be put forward into the interactive discussion and to be sent to referees in its current form. However, I have some suggestions for further improvement, which I recommend to consider during the revision phase in addition to the referees' comments:

1/ Although the "Materials and methods" section is already quite detailed, additional explanations on some important points would be useful:
- Justify the choice of neutron radiography, and explain why this technique is well suited to the problem under investigation.

*We added this in Lines 63-65.*

- Nothing is said about the temperature control during the experiments or the characteristics of the climatic chamber.

*Thank you for pointing this out – we added this in Lines 105-110.*

-You could provide more details on the preparation of the samples and how they are placed in the container. How can you be sure that the samples characterized by X-ray µCT are representative of those in the container (density, SSA, etc.)?

*We added details of the sample preparation in Lines 103-105. Regarding the sample characterization, we did not assume that the snow remained the same after packing it into the column. This is why the density measured with radiography was compared to the µCT density in Table 1. However, characterization of the snow still provided information about the snow samples, though the properties may have changed slightly between the characterization and the experiments. For SSA specifically, since the samples were predominantly melt forms, we do not expect the SSA to have changed significantly.*

-The procedure for the initial filling of the soil, and the control of the hydraulic head during the capillary rise experiments should be better explained. Was the water level in the tank allowed to vary? Was it controlled?

*We improved the description of the sample preparation as mentioned above (Lines 103-105, 108-111).*

-At what time were the capillary rise simulations initialized for the determination of the hydraulic conductivity?

*They were initialized at t=0. We added this to the revised manuscript in Line 226.*

-How exactly are the 1D profiles computed? Along individual lines of pixels? Are they also averaged over some vertical window to integrate the spatial heterogeneity?

*Horizontal averaging was performed for each pixel row. No vertical averaging was performed. We added this in Line 191.*

2/The quality of the fits used to assess the hydraulic conductivity is not discussed. Yet, in some cases, the simulated liquid water content profiles appear to differ significantly from their experimental counterparts. How do these discrepancies affect the reliability of the hydraulic conductivity estimates? Can they explain, at least in part, the large variability of these estimates for the case of gravel samples?

*We added some additional data and discussion regarding the quality of the fits of hydraulic conductivity $K_s$ in Lines 422-433 and added a new figure (Figure A11).*

3/Although the authors deliberately included soil-snow interfaces in their samples, there is only little discussion of the role and effect of these interfaces. Could it be possible that these interfaces induce secondary flows that challenge the assumptions made in the hydraulic modelling?

*The role of the transitional layers below the snow is discussed with respect to the flow rate calculation in Sections 3.2 and 4.1, as well as when discussing the hydraulic conductivity fitting in Section 4.3. The use of 1D fitting is certainly a limitation and we discuss this extensively throughout the manuscript. 2D analyses would certainly allow us to resolve more complex flow effects and are a topic of future research.*

4/The discussion sometimes errs too much on the speculative side. Try to avoid terms such as "somewhat", "it is likely that", etc., and be as specific as possible.

*Where possible, we made more definitive statements such as in Lines 382, 363, and 386.*

5/In line with the above comment, it seems that some of the conclusions emphasized go beyond what can be reliably demonstrated from the results. For example on line 449 about the hysteresis, whereas this hysteresis was not directly investigated in the study. Similarly, on line 449, I would recommend being more nuanced about the possible use of the parameters determined in the study, especially since the role of the soil-snow interface has not been specifically investigated (cf also comment 3 above).

*We assume the Editor meant line 439 for the comment on hysteresis. While we certainly agree that we did not directly investigate hysteresis, we feel that the comparison we performed with the parameterization by Yamaguchi et al. (2012) still supports our statement that hysteresis should be included in models as discussed in detail in Section 4.2.*

*We rephrased the text in Lines 483-484.*

---

## Author Response (AR2)

We would like to thank the Editor for providing further suggestions regarding the fitting. The Editor's comments are in black, plain text and our responses are in *blue, italicized* text.

Quantifying the quality of the fits using the R2 coefficient is a useful addition. However, a proper definition of this metric would be useful.

*After reviewing the fits and concerns raised by the Editor and Reviewers, we have decided to replace $R^2$ with the mean absolute error (MAE), which has physical units (in this case, liquid water content) and therefore adds some additional information as opposed to $R^2$, which is harder to interpret and may be less meaningful for nonlinear functions. We have added the mathematical definition of the MAE as Equation 11 in the Methods section and replaced the discussion related to $R^2$ with discussion of MAE (Lines 347-358, 415-426, and 478-488).*

Furthermore, while the referees primarily express concern about the inverse fitting of the hydraulic conductivity, most of the R2 values indicated in the paper (e.g. in Fig. 4) relate to the fitting of the water retention curve. I recommend adding R2 values for the different capillary rise curves shown in Fig. 5 (and the similar Figures in the appendix). These R2 values for the inverse fitting are plotted in Fig. A11, but it is not necessarily easy to establish a correspondence with the other figures.

*We have added the MAE values for each curve in the inverse fitting figures (Fig. 5, A8-A10) and the water retention curve fits (Fig. 4, A5-A7). We have also therefore removed Figure A11 from the manuscript, as this information is now found in the individual figures of each fit.*

The responses also state that R2 values have been added to Table 1, but I could not find them in the revised manuscript.

*We think the Editor may have misread the response. In our response to Reviewer 1, we stated, "The errors for each of the fitted parameters are included in Table 1 and we added some metrics for the quality of the fit ($R^2$) to the revised manuscript in Figures 4, A5-A7, A11 as well as accompanying text in Lines 426-432." The errors for the individual fit parameters (e.g. n, alpha, $K_s$) were already in Table 1 and the $R^2$ values were only added to the static water retention curve fits and provided over time for the inverse fitting in the newly added Figure A11.  We did not intend to add $R^2$ to Table 1 since it is already nearly too wide for the page.*

*Since we have now replaced $R^2$ with MAE, Figure A11 was removed and the MAE values for the inverse fitting are now provided for each curve in Figures 5 and A8-A10.*

Regarding the discussion, it would be useful to elaborate further on the level of confidence that can be placed in the parameter values in view of these fitting errors, and on the possible causes of the significant discrepancies observed in certain cases.

Simply letting the "reader interpret the discussion given the limitations" outlined in the manuscript (as mentioned in the response to Reviewer 2) does not meet the expected standard for a scientific paper. At the very least, clear guidance should be provided on how these limitations impact the results, particularly when comparing them to those of other studies from the literature.

*We have updated the discussion using the MAE to provide some additional insights in the fit quality and confidence in the fit parameters in Lines 347-358, 415-426, and 478-488.*

In this regard, the newly added paragraph in Section 4.4 mentions the effect of the "relative location of the fitting nodes with respect to the fluctuations in the measurements". I am not sure what this sentence means. What are these fitting nodes, and how are they chosen? The Methods section does not address this issue. How do the final hydraulic conductivity values depend on this choice?

*We agree this was confusing, partially because we had unintentionally referred to these nodes differently in the Methods section (as "observation nodes"). We have removed this sentence as it was supposed to explain additional consequences of the vertical heterogeneity but did so in a confusing way. We have improved our definition of the nodes in the Methods section (Lines 229-238) and the vertical heterogeneity is discussed in Lines 391-402 and 462-469.*